# Through BabyAI Steps: Understanding and Evaluating Grounded Intelligence in LLMs

## Abstract

Does spatial prediction translate to spatial planning in LLMs? We investigate this question through a controlled experimental test bed using a textual adaptation of the procedurally generated BabyAI grid world. Our Predict-Plan-Decompose (PPD) framework evaluates three core aspects of grounded intelligence under full observability: (1) predicting action consequences on environment state, (2) generating action sequences to achieve objectives, and (3) decomposing high-level instructions into subgoal sequences. We find a notable dissociation: while most models achieve over 80% accuracy on spatial prediction, their performance drops to below 20% on multi-step planning. This pattern holds across state-of-the-art models that show similar performance on mainstream benchmarks, yet reveal significant disparities in our evaluation. Under Full mission, Partial observability, Interactive (FPI) execution, performance further degrades to 10-12% success rates in the most challenging settings. We provide a standardized evaluation framework with procedural generation enabling assessment across unlimited environment instances, reducing contamination risks while supporting dynamic evaluation through custom BabyAIBots and virtual environment execution.

## 1 Introduction

To what extent do Large Language Models (LLMs) exhibit grounded intelligence abilities? This fundamental question touches on whether these systems truly understand the world they describe or merely manipulate linguistic patterns. While LLMs demonstrate impressive capabilities in language understanding and generation (Brown et al., 2020; Wei et al., 2022), their capacity for *grounded intelligence* – the ability to align internal representations with environmental dynamics and generate effective action sequences – remains poorly understood (Bisk et al., 2020).

Existing benchmarks often test LLMs on abstract reasoning (Hendrycks et al., 2021a) or evaluate embodied agents in visually complex simulations (Savva et al., 2019; Shridhar et al., 2020) where multiple confounding factors make it difficult to isolate core competencies. However, there is a need for benchmarks specifically designed to probe the planning and reasoning capabilities of LLMs when interacting through language within a procedurally complex, yet controlled, environment where the grounding challenges related to spatial dynamics and object interactions are central.

To address this gap, we introduce **BabyBench**, a suite focused on evaluating grounded reasoning and planning in LLMs by conducting controlled investigations using a textual adaptation of the procedurally generated BabyAI grid world (Chevalier-Boisvert et al., 2019). This underlying platform offers procedurally generated grid-world tasks that demand navigation, object interaction such as manipulating keys, doors, boxes, and understanding of compositional instructions – challenges that are presented to the LLM entirely through text. A key advantage of building upon BabyAI is its flexibility and extensibility; its procedural generation capabilities and systematic level structure allow researchers to easily create new tasks and control difficulty, enabling future expansions.

This experimental test bed allows us to evaluate LLM grounded intelligence through a **Predict-Plan-Decompose (PPD) framework** under full observability, where simplified language and clear symbolic representations create a controlled evaluation laboratory that can reveal fundamental limitations hidden in more complex benchmarks. We further examine these limitations through **Full mission, Partial observability, Interactive (FPI)** experiments that test models under more realistic embodied agent conditions, where the disconnect between predictive and planning capabilities

becomes even more pronounced, with success rates dropping to near-zero on challenging tasks. Surprisingly, we find that visual input degrades performance further: when evaluated with visual environment representations, models achieve near-zero success rates across all tasks, suggesting that current Vision-Language Models default to coarse, region-based spatial representations unsuitable for precise planning.

The core contributions of this work are:

- **Controlled Experimental Framework:** A systematic Predict-Plan-Decompose (PPD) evaluation approach using procedurally generated environments that isolates core grounded reasoning abilities while minimizing confounding factors, complemented by Full mission, Partial observability, Interactive (FPI) experiments that test models under realistic embodied agent conditions (Sections 3.4 and 4.3).
- **Open-Source Evaluation Suite:** Comprehensive evaluation harness providing standardized data generation, procedural environment creation for unlimited instances, interfaces for various LLMs and custom BabyAIBots, comprehensive metrics, and dynamic virtual environment execution for reproducible benchmarking (Section 3.5).[1]
- **Empirical Insights:** Documentation of a fundamental dissociation between spatial prediction and planning capabilities in LLMs, revealing that predictive accuracy does not translate to planning competence, with performance further degrading to near-zero under partial observability constraints (Section 4).

Figure 1 provides a conceptual overview of our PPD framework. Our findings suggest that current approaches to developing spatial reasoning in LLMs may need fundamental reconsideration, as predictive capabilities alone appear insufficient for effective environmental interaction.

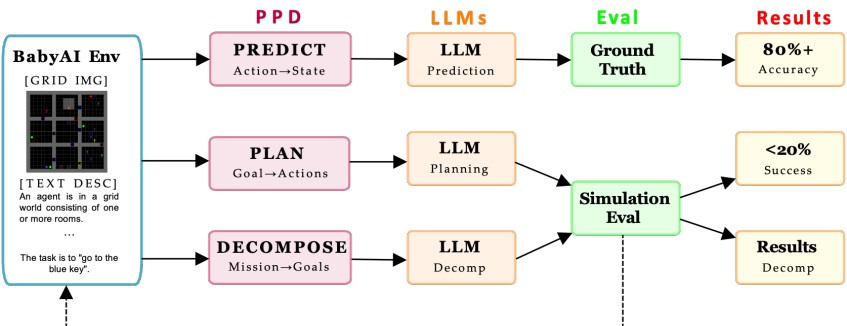

Figure 1: **BABYBENCH PPD Framework Overview.**

## 2 RELATED WORK

LLMs have demonstrated strong performance on a range of reasoning tasks, from arithmetic problems (e.g., GSM8K (Cobbe et al., 2021)) to abstract pattern completion (e.g., Abstraction and Reasoning Corpus (ARC; Chollet, 2019)). However, many of these benchmarks evaluate reasoning in purely symbolic or textual domains, often detached from grounded environments that require spatial understanding, memory, or interaction - fields in which even the most sophisticated models continue to fail (Cai et al., 2025).

There are benchmarks aimed at evaluating models' spatial understanding and grounded reasoning. Approaches like ALFRED (Shridhar et al., 2020) and ALFWorld (Shridhar et al., 2021) take steps in this direction but rely on natural language instructions and/or complex 3D simulation environments. Other grounded environments provide more structured tasks. AI2-THOR (Kolve et al., 2017) and Habitat (Savva et al., 2019) offer visually grounded environments, with extensions such as ProcTHOR (Deitke et al., 2022), ManipulaTHOR (Ehsani et al., 2021), and Habitat 2.0 (Szot et al., 2021). In parallel, purely textual environments like TextWorld (Côté et al., 2018), Jericho (Hausknecht et al., 2020), and ScienceWorld (Wang et al., 2022) evaluate an agent's ability to interpret and act on

---

[1] https://anonymous.4open.science/r/babybench-0166

language instructions without visual input. More recent efforts, such as MineDojo (Fan et al., 2022) and ComplexWorld (Basavatia et al., 2023), integrate open-ended interactive settings with language-defined goals. While these benchmarks offer important insights, they often lack standardization and introduce confounding factors, making it difficult to isolate reasoning from other capabilities. For example, failures in ALFWorld may stem from multiple sources, such as instruction interpretation, spatial reasoning, or action execution. This motivates the need for a controlled evaluation setup designed to disentangle reasoning components. To this end, we propose **BABYBENCH, a benchmark that provides a controlled environment** with clear task definitions and consistent evaluation protocols. Its goal is to **isolate reasoning skills from other confounding factors**, making it possible to evaluate different components independently.

Among these reasoning components, task planning and decomposition are central to solving complex tasks (Gu et al., 2024). In recognition of this, assessing the planning capabilities of LLMs has become an increasingly important objective, motivating the development of benchmarks that test models' ability to generate, adapt, and execute plans in complex scenarios (Irpan et al., 2022; Liang et al., 2023). For example, PLANET (Li et al., 2025b) introduces a broad set of planning tasks spanning embodied environments, web navigation, scheduling, games, and everyday automation. NATURAL PLAN (Zheng et al., 2024) focuses on realistic natural language planning tasks such as trip planning, meeting coordination, and calendar management. In contrast, **BABYBENCH offers a complementary setup**: instead of targeting open-ended planning or instruction following, it evaluates reasoning through interaction with a grounded environment governed by explicit rules and state transitions. Unlike prior datasets that emphasize trajectory imitation or static procedure understanding, **BABYBENCH enables fine-grained diagnosis of where models fail**: whether in planning, decomposition, or answer derivation.

A detailed comparison of BABYBENCH and other benchmarks on the related topics, as well as a further discussion of related work is given in Section B.

## 3 BABYBENCH

### 3.1 BABYAI PLATFORM

BabyAI (Chevalier-Boisvert et al., 2019) is a research platform for studying grounded language learning in language-conditioned agents. Built on MiniGrid (Chevalier-Boisvert et al., 2023), it provides a 2D gridworld where agents interact with objects through 6 discrete actions (`forward`, `right`, `left`, `pick-up`, `drop`, `open`). The platform uses "Baby Language", a compositional subset of English, for instructions like "go to the red door" or "pick up the blue key after you open the yellow door."

Tasks span 19 levels of increasing complexity, with a hand-coded expert bot that decomposes instructions into predefined subgoals (`GoNextToSubgoal`, `OpenSubgoal`, `DropSubgoal`, `PickupSubgoal`, `ExploreSubgoal`). For BABYBENCH, we selected 16 levels that are reliably solvable under our full observability text-based adaptation, excluding configurations that lead to unsolvable scenarios. These levels are organized into four difficulty categories (Easy, Medium, Hard, Very Hard) based on required skills. Full platform details, level descriptions, and examples of excluded configurations are provided in Appendix C.1.

### 3.2 A TEXTUAL ENVIRONMENT

We use the BabyAI environment to evaluate whether LLMs can solve planning tasks. To do this, we convert the environment's state into textual descriptions that are provided as input to the LLM. Prior work (Puerto et al., 2024; He et al., 2024) has shown that prompt formatting can significantly influence LLM performance on reasoning tasks. Therefore, we explore three formatting strategies: (1) *Narrative format*, where the environment is described using natural English sentences; (2) *Structured format*, which combines a sentence-based context with key information presented as bullet-pointed {key:value} pairs; and (3) *JSON format*, which begins with a sentence-based context followed by the remaining information structured in JSON. In Appendix C.7.1, we perform a comparison between the formatting styles). Given the superiority of the ***Structured format*** in terms of metrics described in Section 3.5, **we use this formatter** in the published dataset and in our evaluations of Section 4. Our provided code can easily be adapted to generate data with different formatters.

The evaluated LLM can take on different perspectives depending on the prompt content and ordering. In our benchmark, we adopt the **Omniscient Perspective**, where the model is provided with all available environment information. The task is to solve the problem using this complete context. This setup approximates a human solving the BabyAI environment with full access to state information, ignoring agent-specific constraints such as partial observability. This abstraction moves away from studying perception under limited views, which was a focus in the original BabyAI for analyzing the sample efficiency of RL/IL agents.

### 3.3 BABYAI OMNIBOT

To establish a fair comparison framework for evaluating LLM responses under our full observability setting, we developed an enhanced version of the original BabyAI expert bot called "BabyAI OmniBot." Unlike the original bot that operated under partial observability constraints, the OmniBot leverages complete grid visibility to consistently select optimal navigation paths and implements a revised obstacle classification system. The bot supports two initialization modes: using predefined subgoal sequences or replicating default BabyAI Bot behavior with dynamic subgoal generation. The OmniBot achieves a 100% solve rate across all benchmark levels, providing a robust upper bound for evaluating LLM performance. Full implementation details are provided in Appendix C.5.

### 3.4 TASK DESCRIPTIONS

To assess the spatial reasoning and planning capabilities of language models within a structured environment, we introduce a suite of three complementary tasks named **Predict**, **Plan**, and **Decompose**.

#### 3.4.1 PREDICT: FINAL STATE ESTIMATION FROM ACTION SEQUENCES

The **Predict** task evaluates a model's ability to simulate spatial transformations resulting from a sequence of discrete actions. Specifically, the language model is provided with (i) a textual description of the initial environment, detailing its layout and the objects within it, formatted using one of the strategies described in Section 3.2, (ii) the agent's specific initial position and orientation, and (iii) a sequence of actions (e.g., `left`, `forward`, `pick-up`). The model is then tasked with predicting the final state after executing these actions in order. No interaction with the environment is allowed at inference time; the model must rely entirely on its internal representation of state dynamics. Success in this task requires robust spatial reasoning, such as understanding how orientation changes affect motion, maintaining an internal representation of agent positions over multiple steps, and accounting for constraints in the environment. This task thus serves as a diagnostic for whether the language model has learned a sufficiently detailed world model to reason about transitions deterministically.

The datasets for this task contain tuples of the form: `(level_name, seed, env_description, initial_state, action_sequence, target_state)`.

#### 3.4.2 PLAN: GENERATING ACTION SEQUENCES TOWARD SUBGOALS

The **Plan** task evaluates the model's capacity to synthesize valid action sequences that accomplish a specified subgoal. For this task, the target subgoal is restricted to navigation goals, specifically the `GoNextToSubgoal` subgoal. This design choice focuses the evaluation on the model's ability to plan multi-step movement sequences within the environment. Unlike manipulation subgoals (such as `OpenSubgoal`, `PickupSubgoal`, or `DropSubgoal`), which typically require only a single action once the agent is correctly positioned and oriented, `GoNextToSubgoal` necessitates a sequence of navigation actions potentially involving turns, movement, and avoiding obstacles. Appendix C.3 give more details about the environment for this task.

Given an initial state description and a target `GoNextToSubgoal` argument, the model is tasked with generating a sequence of actions that lead to successful completion of the subgoal when executed in the environment. The correctness of the generated sequence is verified by executing it in the environment. This environmental evaluation is necessary because there is typically no single unique sequence of actions that achieves the subgoal; multiple valid paths and action sequences may exist.

For evaluation, we consider sequences that successfully achieve the subgoal, and among valid sequences, shorter ones are preferred. This task specifically evaluates the model's grounded action

planning ability and implicitly tests its capacity to integrate the environment's transition rules to avoid redundant or non-progressive steps.

The datasets for this task consist of tuples (`level_name, seed, env_description, initial_state, target_subgoal, expert_action_sequence`), where models are tasked with generating an action sequence to achieve the subgoal, and the expert sequence is given for reference only, given the multiplicity of valid action sequences.

### 3.4.3 DECOMPOSE: ABSTRACTION AND INTERMEDIATE GOAL GENERATION

In the **Decompose** task, the model is presented with an initial state and a final high-level goal description, and it must generate a sequence of intermediate subgoals (drawn from a predefined vocabulary of abstract goals introduced in Section 3.1, namely `GoNextToSubgoal`, `OpenSubgoal`, `DropSubgoal`, `PickupSubgoal` - `ExploreSubgoal` being excluded due to the full observability) that guides the agent from the initial state to the final objective. The resulting subgoal sequence is intended to decompose a potentially complex task into smaller, tractable steps. The correctness of the predicted sequence is verified by executing it in the environment. Among multiple valid plans, the shortest ones (in terms of number of actions) are considered optimal. This task tests hierarchical reasoning and goal abstraction: the model must not only understand the structure of the environment and the agent's capabilities but also break down a long-horizon objective into a sequence of logically proximate tasks.

The datasets for this task consist of tuples (`level_name, seed, env_description, initial_state, mission, help_count`). Here, the help_count is the number of subgoals added by the OmniBot with default initialization (see Section 4.2.3).
The evaluation process is explained in Appendix F.4.2.

### 3.5 EVALUATION METRICS

We evaluate model performance by comparing LLM outputs against ground truth derived from or executred by the **BabyAI OmniBot**, which serves as a strong symbolic baseline operating with full environment visibility, as described in Section 3.3. The task-specific metrics are summarized in Table 1, and are detailed in Table F.1 and Appendix F.

Table 1: Summary of Evaluation Metrics for BABYBENCH tasks

| Task | Metric | Description |
|---|---|---|
| **Predict** | **Success Rate** | Proportion of correct final state predictions. |
| | **Manhattan Distance** | L1 distance between predicted and correct agent position (for incorrect predictions). |
| **Plan** | **Success Rate** | Proportion of tasks successfully completed by executing LLM-generated action sequences. |
| | **Efficiency Ratio** | Ratio of the length of OmniBot's optimal actions sequence to LLM's actions sequence for successful plans. |
| **Decompose** | **Comprehension Rate** | Success rate when OmniBot executes LLM subgoals, allowing additions. |
| | **Precision Rate** | Success rate when OmniBot executes *only* LLM subgoals (no additions). |
| | **Assistance Curve Integral (ACI)** | Area under the curve of success rate vs. number of allowed additional bot subgoals ($k$). |

## 4 BASELINE EVALUATIONS

### 4.1 EXPERIMENTAL CONFIGURATION

**Used models**: To assess our proposed benchmark, we use a set of cutting-edge LLMs. This selection includes Claude 4 Sonnet (Anthropic, 2025), known for its strong reasoning and coding capabilities. We also evaluate Meta's Llama 3.1 (Grattafiori et al., 2024) across its diverse range of parameter sizes (specifically, the 8B, 70B, and the notably large 405B versions) to understand performance scaling. We also include Llama-3.1 70B Distilled from DeepSeek (DeepSeek-AI et al., 2025), referred further as "DeepSeek-R1-Distill and DeepSeek-R1". Furthermore, we incorporate Qwen3 32B (Team, 2025), a model featuring advanced reasoning and multilingual support. Finally, our evaluation includes

GPT-5 (OpenAI, 2024), a flagship multimodal model well known for its strong performance on a wide array of AI tasks. This comprehensive set of models allows for a thorough examination of state-of-the-art models on our benchmark. We use DeepInfra[2] for model inference, and also integrate API-based access to Claude and GPT-5 to cover these models as well.

**Evaluation Setting**: We evaluate the LLMs under various prompting strategies to assess their reasoning capabilities. This includes: zero-shot evaluations, where models perform tasks without any specific examples; few-shot learning (Brown et al., 2020), where models are provided with 3 illustrative examples (BabyAI OmniBot's optimal solutions) within the prompt; Chain-of-Thought (CoT) prompting (Wei et al., 2022), which encourages models to generate explicit step-by-step reasoning before arriving at an answer; and Tree-of-Thought (ToT) prompting (Yao et al., 2023), which allows models to explore and evaluate multiple reasoning paths for more complex problem-solving.

Initial investigation described in Appendix E indicated the superiority of the **ToT prompting strategy, which we adopt for all experiments in this section**.

## 4.2 PERFORMANCE ANALYSIS

### 4.2.1 PREDICT

We report the results of the **Predict** task in Table 2, with task difficulties defined in Table 5 (Appendix C.2). Among all models evaluated, Claude demonstrates the strongest performance, correctly predicting both position and direction in more than 80% of the samples, even on the most challenging levels. In contrast, while GPT-5, DeepSeek-R1-Distilled model perform comparably to Claude on easier levels, their accuracy drops significantly as difficulty increases. The three LLaMA models exhibit the expected trend of improved performance with increasing model size.

Table 2: Average Accuracy Across Task Difficulty Levels (%)

| Model | Easy | Medium | Hard | Very Hard |
|---|---|---|---|---|
| Claude | 100.00 | 94.30 | 85.00 | 81.20 |
| DeepSeek | 94.30 | 73.40 | 58.80 | 38.80 |
| GPT-5 | 97.50 | 92.80 | 82.70 | 72.70 |
| Llama-405B | 97.50 | 87.90 | 81.70 | 61.30 |
| Llama-70B | 56.40 | 49.20 | 49.00 | 45.90 |
| Llama-8B | 5.00 | 4.70 | 6.00 | 1.90 |
| Qwen | 86.80 | 73.60 | 63.20 | 38.90 |

Competence-wise evaluation shows that Claude achieves the highest success rates across all competency categories, ranging from 76.7% to 99.2%, with particularly strong room navigation performance (99.2%). Other models exhibit substantial degradation in complex scenarios: while DeepSeek-R1-Distill, GPT-5, and Qwen perform reasonably in single-room environments, their accuracy drops markedly in maze navigation. Sequential instruction processing proves most challenging, with all models except Claude showing significant difficulties. Qualitative analysis reveals that Claude maintains systematic step-by-step reasoning, whereas other models tend to deviate from tasks or question problem setups rather than executing methodical solutions. A more detailed analysis of this is given in Section G.

### 4.2.2 PLAN

We evaluated model performance on a planning task using three grid sizes: small, medium, large, and ultra. The difficulty of the task scaled with grid size, primarily due to a corresponding increase in the number of obstacles. We used small grids (room size of 8, up to 7 obstacles), medium grids (room size of 16, up to 60 obstacles), large grids (room size of 24, up to 120 obstacles), ultra grids (room size of 32, up to 180 obstacles). Then we run all models on 5 seeds within these settings and measured the resulting success rate.

As illustrated in Figure 2, all evaluated models demonstrated some capability of solving the task. At least one successful solution was provided for small and medium grids by each model, for large and

---

[2]https://deepinfra.com/

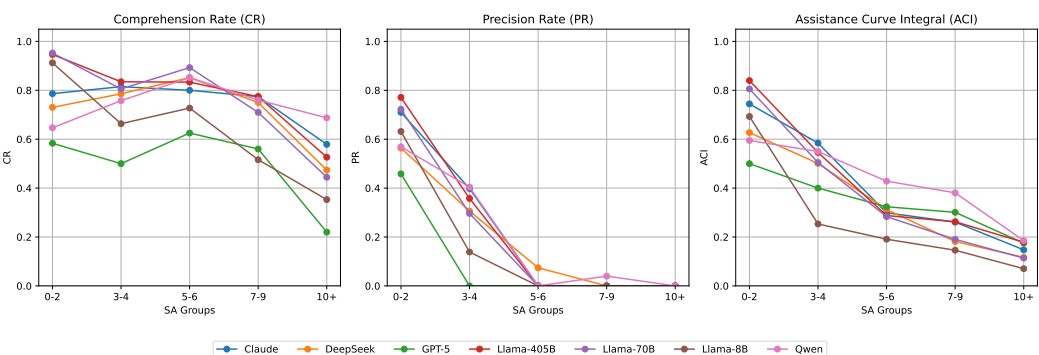

Figure 3: CR, PR and ACI for all models, for different classes of missions on the decompose task.

ultra grids only smaller Llama models failed completely. However, Llama models failed to achieve success rate of above 20% even on the small grid. Reasoning models, in contrast, perform well on small grids, achieving over 70% success rate (Claude, and GPT-5). Medium and large grids still present a challenge for these models, since none of them could beat the 50% success rate. Ultra large grids are even more challenging, only the reasoning models (and Llama 405B) are capable of scoring over 0 on them, and the maximum performance is about 20% (Claude). It illustrates that our benchmark is challenging for the models.

Detailed analysis of the Plan task, including overall success rate and model efficiency, as well as the specific impact of the number of obstacles on performance, are presented in Appendices H.2 and H.1.

### 4.2.3 DECOMPOSE

For each mission, we find how many subgoals are added by the OmniBot when initialized with a subgoal stack that contains only a direct translation of the task description (see Appendix C.6 for an example of the evolution of the stack of subgoals during the execution of one episode). To assess the effectiveness of LLMs on the decomposition task, we separate missions

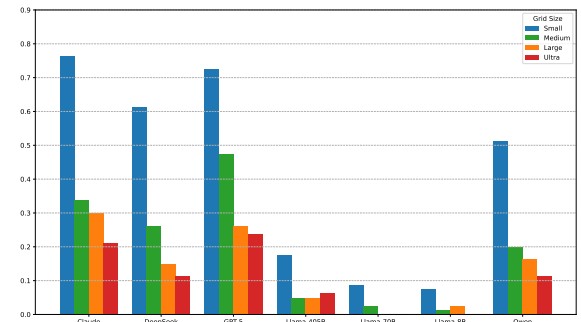

Figure 2: Success Rate of the Models on Varying Grid Sizes for the Plan task.

into different classes depending on this quantity, to which we will refer as SA (Subgoals Added) in this subsection for readability. We interpret SA as the difficulty of the mission: a mission is hard when the OmniBot needs to add many intermediary subgoals to solve it. Figure 3 shows the results obtained for all models and the three metrics.

We observe a substantial degradation in performance as SA increases, particularly evident for PR. This pattern has a straightforward explanation. For missions with 0-2 SA, achieving high PR generally requires only translating the mission to a subgoal. For example, if the mission is `go to the red ball` and SA is zero, LLMs can simply output `GoNextToSubgoal((x,y))`, where `(x,y)` are the coordinates of the red ball in the grid that were explicitly provided to the LLM. However, for 7-9 SA, only Qwen achieve non-zero PR, and for 10+ SA, all models have PR=0. This demonstrates that current LLMs cannot effectively decompose long-horizon missions into elementary subgoals while accounting for the agent's operational environment. The declining trend of CR and ACI confirms that as mission complexity increases, LLMs struggle to integrate available information and generate coherent subgoals.

### 4.3 FPI EXPERIMENTS

To complement our claim, we introduce FPI (Full mission, Partial observability, Interactive) experiments that test LLMs under more realistic embodied agent conditions by requiring full mission completion with limited environmental information based on the agent's field of view and step-by-step interactive execution.

Using the same 16 BabyAI levels across four difficulty categories with varying agent view sizes (3, 5, 7, 9), we find that both Claude-Sonnet-4 and GPT-5 achieve strong performance on Easy tasks (59-74% success rates), but experience substantial degradation as complexity increases, dropping to approximately 30% on Moderate tasks and 36-43% on Hard tasks. Most strikingly, both models struggle significantly on Very Hard tasks, achieving only 10-12% success rates, indicating that the combination of long-horizon planning, partial observability, and interactive execution creates a substantial challenge for current LLMs, with GPT-5 maintaining slightly better efficiency (7-9%) compared to Claude-Sonnet (4%) on the most challenging configurations. More details on the FPI experiments are in Appendix D.

Table 3: Experimental Results: Success Rate / Efficiency (%)

|  | Easy | | Moderate | | Hard | | Very Hard | |
|---|---|---|---|---|---|---|---|---|
|  | 3 | 5 | 5 | 7 | 5 | 7 | 7 | 9 |
| Claude-Sonnet-4 | 74.0 / 60.0 | 73.7 / 47.1 | 31.0 / 27.0 | 34.0 / 32.1 | 36.7 / 30.2 | 40.7 / 31.1 | 10.2 / 4.6 | 11.7 / 4.0 |
| GPT-5 | 59.0 / 56.4 | 74.0 / 58.2 | 29.0 / 26.5 | 32.0 / 43.2 | 43.3 / 33.5 | 43.3 / 38.8 | 11.7 / 9.1 | 11.7 / 7.1 |

## 5 ANALYSIS

### 5.1 THE PREDICT–PLAN ASSYMETRY

We observe a clear asymmetry across tasks. Several language models accurately simulate a given action sequence, reliably returning the agent's final coordinates. The same models, however, underperform when asked to generate an action sequence that reaches specified coordinates, and they likewise struggle to decompose a goal into valid, structured subgoals. This gap persists across prompts and instances, yielding high scores on forward state prediction but markedly lower scores on planning and hierarchical decomposition.

We believe this arises for two main reasons. The first and most straightforward reason is that **Predict** is essentially Markovian: the outcome of the action at time t depends only on the agent's current state. As a result, **Predict** closely mirrors the pretraining objective of LLMs, namely next-token prediction. By contrast, applying the same logic to **Plan** induces a greedy, myopic search procedure, which unsurprisingly leads to poor performance.

The second reason is that **Predict** does not require the LLM to form a faithful spatial internal representation. In practice, models that perform well on this task implement a shallow arithmetic routine rather than reasoning about the environment. As a result, the model can ignore the spatial context and rely solely on integer additions (see Appendix G.1). By contrast, LLMs that failed on **Predict** tended to over-engage with the spatial context, for example by validating each action, checking the prompt, or questioning feasibility. These behaviors introduce error modes that are irrelevant to the arithmetic core of the task. The poor performance of models that succeed on **Predict** when evaluated on **Plan** is also expected, since planning cannot be done reliably without interacting with the environment to assess feasibility and enumerate admissible next actions. Taken together, these observations indicate that current LLMs still struggle to build and exploit robust spatial representations, even in small static grids with discretized objects, which is at odds with the demands of real-world embodied settings.

### 5.2 BEHAVIORS CAUSING FAILURE

We observed that the Qwen model often gets caught in a "labyrinth of self-correction," ultimately failing due to analysis paralysis (an example of Qwen reasoning is given in Section I.3). In contrast, models like Meta-Llama3.1-405B, GPT, and Claude tend to produce structured, markdown-based reasoning (see Section I.1). For Predict tasks, the main source of failure across models was an inability to understand the environment's logic: they frequently confuse turns, directions, or coordinates, leading to navigation errors. In Plan tasks, models often overlook obstacles, resulting in simplified plans that cannot be executed. In Decompose tasks, models sometimes fail to track the state of objects (e.g., attempting to find a key for an already open door) or hallucinate obstacles that

do not exist. Overall, these issues are primarily linked to spatial reasoning. The additional constraints in the FPI setting further accentuate these behaviors.

### 5.3 TEXT DESCRIPTION VS. VISUAL DESCRIPTION

We conducted experiments with GPT5 using two modalities for specifying the environment: a purely textual description and a visual description (a screenshot of the initial state). Across tasks, the Vision-Language Model (VLM) variant of GPT5 consistently achieved *worse* results than its LLM counterpart (these results are not reported since they yielded zero outcomes). We attribute this gap to systematic differences in how the two models reason about the environment.

The VLM tends to ground its plan in coarse, imprecise spatial cues. Its descriptions of obstacles are vague, and even the main objective is localized only approximately (e.g., "The red ball is in the lower middle-left area of the grid."). Before committing to a concrete sequence of actions, it produces underspecified directives such as "Move west a few tiles" or "Make a short south adjustment." Such language indicates that the internal representation remains qualitative rather than metric: positions are framed as regions, movements as tendencies, and obstacle layouts as fuzzy patterns. This imprecision propagates into planning, yielding action sequences that are only loosely constrained and therefore more error-prone when precise coordination is required. This behavior also accounts for the complete failure of GPT5-VISION on the *Predict* task: it does not consistently track the agent's coordinates through to the end of the action sequence. Even when the input trajectories are pre-validated to be fully feasible and obstacle-free, the model hallucinates intervening walls. We attribute this to its coarse, region-level spatial representation, which permits spurious barrier inferences and prevents accurate state propagation across successive actions.

By contrast, the LLM instantiates a more discrete, symbolically precise reasoning style. It typically begins by specifying the exact location of the goal (e.g., "the red ball is located at cell $(4, 14)$"), then identifies the agent's initial state, and proceeds to select and evaluate candidate moves step by step while explicitly accounting for nearby obstacles. Although this procedure does not eliminate all errors, it consistently reduces compounding mistakes and improves overall performance.

## 6 CONCLUSION

BABYBENCH offers a controlled, text-based evaluation of grounded reasoning across three tasks (**Predict**, **Plan**, and **Decompose**) with a standardized, success-rate–based harness. Empirically, we find a persistent *Predict–Plan asymmetry*: models that accurately roll out states often fail at goal-conditioned planning and hierarchical decomposition. Visual conditioning (e.g., GPT5-VISION) surprisingly underperforms text, achieving near-zero success rates due to coarse, non-metric spatial representations. Trace audits further reveal systematic errors in orientation handling, obstacle awareness, and object-state tracking. Together, these results expose concrete gaps in spatial reasoning and planning, while the benchmark's reproducible protocol and extensibility make it a practical yardstick for future methods.

## 7 LIMITATIONS

Our work introduces BABYBENCH, a benchmark suite for evaluating grounded planning and reasoning in LLMs, and makes several design choices that imply certain limitations. While we evaluate a diverse set of state-of-the-art LLMs to demonstrate the benchmark's capabilities, we do not provide an exhaustive empirical study of all available models or explore finetuning strategies. Additionally, our evaluation primarily focuses on full observability settings (PPD), with FPI experiments providing only initial insights into partial observability challenges. The benchmark uses discrete grid worlds with symbolic objects, which may not capture all complexities of continuous environments. Finally, while we identify the surprising failure of vision-language models, we evaluate only one VLM (GPT-5), and further investigation with other multimodal models would strengthen these findings.

## REPRODUCIBILITY STATEMENT

To ensure full reproducibility of our results, we provide comprehensive implementation details and make all necessary resources publicly available. The complete source code for BABYBENCH, including dataset generation, evaluation harness, and baseline experiments, is available at `https://anonymous.4open.science/r/babybench-0166`.

Our implementation includes: (1) procedural dataset generation scripts for creating unlimited instances of Predict, Plan, and Decompose tasks as described in Section 3.4, (2) the BabyAI OmniBot implementation detailed in Section 3.3 that serves as our expert baseline, (3) textual environment formatters (narrative, structured, JSON) discussed in Section 3.2 and Appendix C.7.1, (4) complete evaluation metrics implementation as specified in Section 3.5 and Table 1, and (5) interfaces for all evaluated LLMs including API-based models (Claude, GPT-5) and open models via DeepInfra.

All hyperparameters and experimental configurations are documented in Section 4.1, with default values provided in the codebase. The repository includes pre-generated datasets in the datasets folder with clear descriptions, though researchers can generate new instances using scripts in the generate folder to avoid potential contamination. Setup instructions specify exact dependencies including MiniGrid commit hash to ensure environment consistency.

The evaluation pipeline supports both static dataset evaluation and dynamic environment-based execution for Plan and Decompose tasks. We provide scripts for reproducing all experimental results reported in Section 4, including performance analysis across difficulty levels (Table 2) and competency-based evaluation (Table 9). The modular design allows researchers to easily extend the benchmark with new BabyAI levels, formatting styles, or evaluation models beyond the 16 levels and 7 models presented in this work.

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

## A    LLM USAGE

We used LLMs to aid and polish writing throughout this manuscript. Specifically, we employed LLMs for grammatical refinement, clarity improvements, and ensuring consistency in technical terminology across sections. All scientific content, experimental design, results, and interpretations represent original work by the authors, with LLMs serving solely as writing assistants for language enhancement.

## B    DETAILED RELATED WORK

Table 4: Comparison of BABYBENCH (Ours) with related grounded reasoning benchmarks. ✓ indicates full support, ▲ indicates partial or limited support, and ✗ indicates no support.

| Feature | BABYBENCH | ARC | ALFRED | ALFWorld | GTB |
|---|---|---|---|---|---|
| *Reasoning Skills* | | | | | |
| Multi-step planning | ✓ | ✗ | ✓ | ✓ | ✓ |
| Spatial reasoning | ✓ | ✓ | ✓ | ✓ | ✓ |
| Hierarchical planning | ✓ | ✗ | ✓ | ✓ | ✗ |
| Goal decomposition | ✓ | ▲ | ✓ | ✓ | ✗ |
| Predictive reasoning (action effects) | ✓ | ▲ | ▲ | ▲ | ▲ |
| *Environment and Evaluation* | | | | | |
| Interactive environment feedback | ✓ | ▲ | ✓ | ✓ | ✓ |
| Traceability of reasoning steps | ✓ | ✗ | ▲ | ▲ | ▲ |
| Subtask supervision | ✓ | ✗ | ✓ | ✓ | ✗ |
| Text-based interface | ✓ | ✗ | ✗ | ✓ | ✓ |
| Symbolic world state access | ✓ | ✓ | ✗ | ✓ | ✓ |
| *Benchmark Properties* | | | | | |
| Procedural task generation | ✓ | ✗ | ✓ | ✓ | ✓ |
| Difficulty scaling | ✓ | ✗ | ▲ | ▲ | ▲ |
| Ground-truth plans available | ✓ | ✗ | ✓ | ✓ | ✓ |
| Supports diverse LLMs (open/closed) | ✓ | ▲ | ✓ | ✓ | ✓ |
| Contamination Free | ✓ | ✓ | ✗ | ✗ | ✓ |
| Reproducibility | ✓ | ✓ | ✓ | ✓ | ✓ |

BABYBENCH bridges multiple research areas, including benchmarks for grounded environments, reasoning and planning evaluation in LLMs, procedural understanding datasets, and the use of expert demonstrations.

**Grounded Agent Benchmarks**    Evaluating agents in interactive environments requires benchmarks that provide structured tasks and controlled settings. Prior work has explored visually grounded environments such as AI2-THOR (Kolve et al., 2017) and Habitat (Savva et al., 2019), along with extensions such as ProcTHOR (Deitke et al., 2022), ManipulaTHOR (Ehsani et al., 2021), and Habitat 2.0 (Szot et al., 2021). Other work has focused on language-based interaction in purely textual environments. For instance, TextWorld (Côté et al., 2018) is based on classic adventure games, Jericho (Hausknecht et al., 2020) focuses on interactive fiction, and ScienceWorld (Wang et al., 2022) introduces tasks requiring scientific reasoning. These settings evaluate an agent's ability to interpret and act on language instructions without visual input. Recent work has begun to explore LLM agents in interactive environments. For example, MineDojo (Fan et al., 2022) integrates Minecraft with language-defined goals, while ComplexWorld (Basavatia et al., 2023) introduces a text-based reinforcement learning environment for simulating open-ended quests.

While prior benchmarks offer useful insights, they often lack standardization or introduce additional complexities that make it difficult to isolate and measure core reasoning capabilities. Recent work has applied LLMs to downstream settings such as robot control (Huang et al., 2022), and reinforcement learning with grounded feedback (Carta et al., 2023). These advances highlight the need for a

benchmark that is both simple and standardized, enabling focused evaluation of reasoning skills without confounding factors. **To this end, we propose BABYBENCH, which provides a controlled environment with clear task definitions and consistent evaluation protocols.** As shown in Table 4, BABYBENCH not only covers more reasoning skill than other benches, but it also has the most sophisticated evaluation setup. Additionally, since BABYBENCH uses procedural generation of tasks, unlike other benchmarks, it provides quasi infinite number of levels, which makes it unlikely to suffer form contamination.

**LLM Reasoning and Planning Evaluation**   With the rise of LLMs, many benchmarks have been proposed to evaluate their emerging reasoning capabilities. Chain-of-Thought (CoT) prompting (Wei et al., 2022) has shown that LLMs can perform multi-step reasoning, leading to the development of evaluation sets targeting reasoning and planning skills. Research has also explored using methods like preference optimization applied to reasoning traces to enhance model performance on such tasks (Lahlou et al., 2025). General-purpose benchmarks such as Big-Bench Hard (BBH) (Suzgun et al., 2022) and Big-Bench Extra Hard (BBEH) (Kazemi et al., 2025) include a wide range of tasks, from logic puzzles and mathematical proofs to strategy games and commonsense reasoning, providing broad coverage across reasoning domains. In parallel, specialized datasets have been introduced to assess specific modalities, such as mathematical reasoning, GSM8K (Cobbe et al., 2021; Hendrycks et al., 2021b), spatial reasoning (Ma et al., 2024; Rajabi & Kosecka, 2024), financial reasoning (Reddy et al., 2024). Assessing the planning capabilities of LLMs has become an increasingly important objective, motivating the development of benchmarks that test models' ability to generate, adapt, and execute plans in complex scenarios (Irpan et al., 2022; Liang et al., 2023). For example, PLANET (Li et al., 2025b) introduces a broad set of planning tasks spanning embodied environments, web navigation, scheduling, games, and everyday automation. NATURAL PLAN (Zheng et al., 2024) focuses on realistic natural language planning tasks such as trip planning, meeting coordination, and calendar management. **In contrast, BABYBENCH provides a complementary setup where planning and reasoning are evaluated through interaction with a grounded environment governed by explicit rules and state transitions.**

**Procedural Understanding Datasets**   Understanding and generating sequences of steps in dynamic environments is a core component of grounded reasoning. Prior datasets such as WikiHow (Koupaee & Wang, 2018) and RecipeQA (Yagcioglu et al., 2018) provide large-scale procedural text data, often with accompanying images or videos, but they are not embedded in interactive or executable environments. Instruction-following datasets like ALFRED (Shridhar et al., 2020) offer a more grounded setup by pairing high-level goals with action sequences in simulated visual environments. Recent work has aimed to narrow the gap between static procedural text and interactive execution. IndustReal (Schoonbeek et al., 2024) introduces a multimodal dataset focused on step-by-step procedure recognition in industrial settings, with annotations for execution errors and omissions to support robustness analysis. FlaMBé (Dannenfelser et al., 2023) curates biomedical procedures from expert-written documents, supporting structured extraction of complex workflows from unstructured text. **BABYBENCH provides structured input-output pairs that directly target prediction, planning, and decomposition, using expert demonstrations in an interactive simulation. Unlike prior datasets that emphasize full trajectory imitation or static procedure understanding, our setup enables fine-grained evaluation of reasoning components in a controllable environment.**

**Expert Demonstrations Utilization**   Expert data plays a central role in imitation learning (Zare et al., 2024) and offline RL (Levine et al., 2020), and can originate from a variety of sources, including optimal demonstrations by humans and high-quality trajectories produced by scripted agents or bots (Fan et al., 2024). The goal of using such data is typically twofold: to train agents that can imitate expert behavior or to enable learning from past experiences without requiring further environment interaction (Park et al., 2024). Such imitation learning datasets are traditionally composed of raw state-action trajectories. However, learning purely from such sequences often fails to capture the underlying structure, intent, or causal dependencies behind expert decisions. As a result, agents may exhibit shallow imitation: replicating surface-level behavior without acquiring the capacity to generalize to novel scenarios or reason about unseen subgoals (Li et al., 2025a). To address this limitation, our data generation framework builds on the expert bot from Chevalier-Boisvert et al. (2019), ensuring optimal or near-optimal behavior traces. Crucially, we move beyond raw trajectory logging: we decompose expert traces into structured components such as state transitions,

subgoal-aligned action sequences, and hierarchical subgoal plans. These are then used to formulate a suite of benchmark tasks specifically designed to probe distinct reasoning capabilities. **Our dataset can be used for imitation learning, and it differs from conventional imitation learning sources by offering structured, decomposed supervision signals that enable fine-grained evaluation of reasoning, planning, and hierarchical understanding, rather than simply using end-to-end expert behavior.**

## C   MORE DETAILS ON BABYBENCH

### C.1   BABYAI PLATFORM

BabyAI (Chevalier-Boisvert et al., 2019) is a research platform designed to study grounded language learning and sample efficiency in language-conditioned agents. It provides a controlled environment where agents learn to follow language instructions with minimal supervision, similar to human language acquisition.

The BabyAI platform is based on the MiniGrid environment (Chevalier-Boisvert et al., 2023). MiniGrid, which is licensed under the Apache License 2.0, is a partially observable 2D gridworld where agents navigate with a limited field of view and interact with color-coded objects (doors, keys, balls, and boxes) and walls through 6 discrete actions: `forward`, `right`, `left`, `pick-up`, `drop`, `open`. The `right` and `left` actions change the agent's orientation, and the `forward` action changes the agent's coordinate on the grid, as long as it is not facing a wall or an object that is not an open door. The BabyAI platform itself is licensed under the BSD-3-Clause license. Communication within the BabyAI platform occurs through "Baby Language", a compositional subset of English with formal grammar rules, supporting instruction templates such as "go to the red door" or "pick up the blue key after you open the yellow door."

Tasks are organized across 19 levels of increasing complexity, involving navigation, object manipulation, and multi-step goals that require hierarchical reasoning and memory. The platform includes a hand-coded expert bot that can solve these tasks. This bot decomposes complex instructions into predefined subgoals (`GoNextToSubgoal`, `OpenSubgoal`, `DropSubgoal`, `PickupSubgoal`, `ExploreSubgoal`) that take specific object arguments and are executed sequentially, enabling the bot to break down complex instructions into manageable sequences for generating expert demonstrations.

BabyAI supports training agents using both imitation learning, leveraging demonstrations generated by an expert bot (Section 3.3), and reinforcement learning with sparse rewards. The 19 levels are designed with increasing difficulty to facilitate research into curriculum (Srinivasan et al., 2019; Willems et al., 2020), allowing agents to be gradually exposed to more complex tasks and language constructs. The platform's modular and efficient MiniGrid architecture enables rapid experimentation with various learning algorithms and curricula. BabyAI serves as an effective testbed for studying grounded language learning, compositional generalization, and, importantly, the challenges in achieving human-like sample efficiency with current methods, as the paper highlights that significant data is required to solve even seemingly simple tasks.

For BABYBENCH, we selected a subset of the original 19 BabyAI levels. Our selection includes 16 levels that are reliably solvable within our text-based environment adaptation and under the omniscient observation setting provided to the LLM. We found that certain environment configurations generated by the original levels could lead to unsolvable scenarios under these specific conditions, necessitating this filtering. An example of such an unsolvable environment instance that led to the exclusion of certain level configurations is presented in Figure 5 (Appendix C). The 16 levels included in BABYBENCH (GoToObj, GoToRedBallGrey, GoToRedBall, GoToLocal, PutNextLocal, PickupLoc, GoToObjMaze, GoTo, Pickup, UnblockPickup, Open, Synth, SynthLoc, GoToSeq, SynthSeq, and BossLevel), along with the skills they are designed to assess (based on the original BabyAI taxonomy), are summarized in Table 5 (Appendix C.2). They are organized into four difficulty categories: Easy, Medium, Hard, and Very Hard.

## C.2 THE 16 LEVELS OF BABYBENCH

This section details the 16 levels selected from the original BabyAI benchmark for inclusion in the BABYBENCH suite. Table 5 summarizes these levels, grouped by difficulty category (Easy, Medium, Hard, Very Hard), and indicates the specific core skills each level is designed to assess, following the taxonomy presented in the original BabyAI paper (Chevalier-Boisvert et al., 2019).

Table 5: BabyAI Skills Required by Level Difficulty

| Skills | Easy | | Medium | | | | | | | Hard | | | Very Hard | | | |
|---|---|---|---|---|---|---|---|---|---|---|---|---|---|---|---|---|
| | GoToObj | GoToRedBallGrey | GoToRedBall | GoToLocal | PutNextLocal | PickupLoc | GoToObjMaze | GoTo | Pickup | UnblockPickup | Open | Synth | SynthLoc | GoToSeq | SynthSeq | BossLevel |
| Navigate a 6×6 room | ✓ | ✓ | ✓ | ✓ | ✓ | ✓ | ✓ | ✓ | ✓ | ✓ | ✓ | ✓ | ✓ | ✓ | ✓ | ✓ |
| Ignore grey box distractors | | ✓ | ✓ | ✓ | ✓ | ✓ | | ✓ | ✓ | ✓ | ✓ | ✓ | ✓ | ✓ | ✓ | ✓ |
| Ignore all distractors | | | ✓ | ✓ | ✓ | ✓ | | ✓ | ✓ | ✓ | ✓ | ✓ | ✓ | ✓ | ✓ | ✓ |
| Navigate a 3×3 maze of 6×6 rooms | | | | | | | ✓ | ✓ | ✓ | ✓ | ✓ | ✓ | ✓ | ✓ | ✓ | ✓ |
| Move objects out of the way to navigate | | | | | | | | | | ✓ | | ✓ | ✓ | | ✓ | ✓ |
| Unlock doors when explicitly instructed | | | | | | | | | | | | ✓ | ✓ | | ✓ | ✓ |
| Unlock doors when not explicitly stated | | | | | | | | | | | | | | | | ✓ |
| Understand "go to" instructions | | | | ✓ | | | | ✓ | | | | ✓ | ✓ | ✓ | ✓ | ✓ |
| Understand "open" instructions | | | | | | | | | | | ✓ | ✓ | ✓ | | ✓ | ✓ |
| Understand "pick up" instructions | | | | | | ✓ | | | ✓ | ✓ | | ✓ | ✓ | | ✓ | ✓ |
| Understand "put" instructions | | | | | ✓ | | | | | | | ✓ | ✓ | | ✓ | ✓ |
| Understand relative location | | | | | | ✓ | | | | | | | ✓ | | ✓ | ✓ |
| Understand sequences of instructions | | | | | | | | | | | | | | ✓ | ✓ | ✓ |

## C.3 CUSTOM ENVIRONMENT FOR THE PLAN TASK

Since the Plan task exclusively involves `GoNextTo` subgoals, we designed a custom environment derived from the BabyAI level `GoToTheRedBallGrey`. This environment allows for varying grid sizes and configurable numbers of distractor objects (items placed to obstruct the agent's path), requiring it to navigate around them. There are 4 grid sizes : Small (8x8), Medium (16x16), Large (24x24), Ultra (32x32). Figure 4 shows an example of such a grid.
This custom environment will be named `CustomBabyAI-GoToRedBall-{size}-{number of distractors}Dists-v0` depending on the size and the number of distractors.

## C.4 EXAMPLE OF UNSOLVABLE BABYAI ENVIORNMENTS

As discussed in Section 3.1, certain configurations generated by the original BabyAI levels can result in missions that are impossible to complete within our benchmark's setup. Figure 5 shows one such example from an instance of the Unlock level (generated with seed 63). The instruction is to "open the green door". To open a locked green door, the agent requires a green key. However, in this specific environment configuration, the only green key is located inside a room that is locked by a green locked door. Identifying and excluding such unsolvable configurations ensures that the benchmark tasks are always logically coherent and achievable.

## C.5 BABYAI OMNIBOT

The original BabyAI platform included a hand-coded expert called "BabyAI Bot", designed to simulate a human teacher by generating successful demonstrations. Crucially, this bot operated under the same limited observability constraints as the learning agent, only using information it could realistically have access to through exploration.

In our current benchmark, we provide the LLM with full observability of the environment. The environment descriptions reveal the positions of all objects in the grid regardless of the agent's

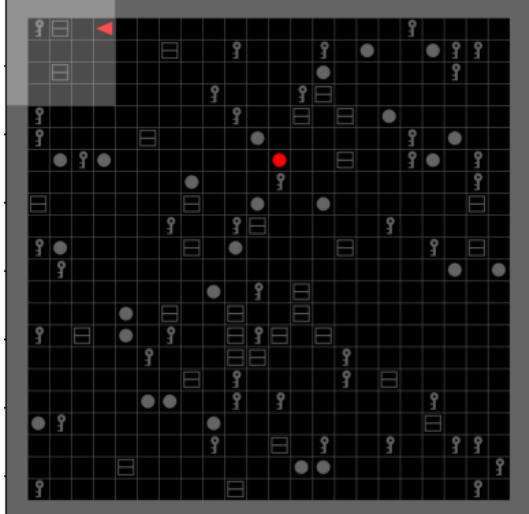

Figure 4: An example of custom environment specifically designed for Plan task, with an 32x32 grid size and 180 objects: **CustomBabyAI-GoToRedBall-Ultra-180Dists-v0**

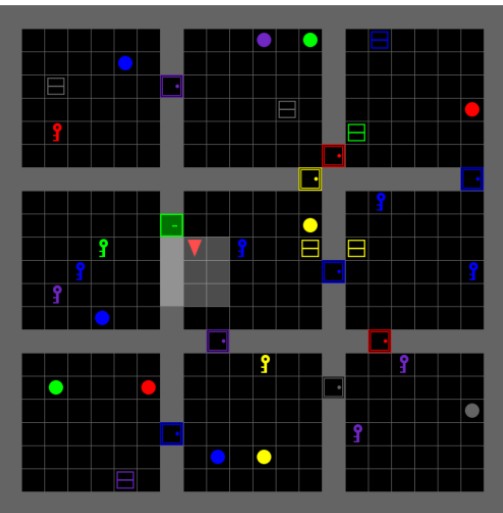

Figure 5: BabyAI Unlock level with seed 63. The mission is to "open the green door". This example illustrates why some levels of the original BabyAI platform are unsolvable, and why they were removed from BABYBENCH.

position or direction. To establish a fair comparison framework for evaluating LLM responses, we developed an enhanced version of the expert called "BabyAI OmniBot."

This new OmniBot retains the core logic of the original bot but has been modified to leverage complete grid visibility. The modifications eliminate the need for exploration as the bot now has full environmental awareness, allowing it to consistently select optimal navigation paths. Additionally, the OmniBot implements a revised obstacle classification system. While the original bot considered unlocked but closed doors as soft obstacles and locked doors as hard obstacles (equivalent to walls), the OmniBot classifies locked doors as soft obstacles since it can immediately locate keys and retrieve them before approaching locked doors.

We extend the bot's initialization capabilities by supporting two distinct modes:

- Predefined subgoal sequence: The bot can be initialized with a sequence of predefined subgoals that it must follow. It retains the flexibility to append additional subgoals dynamically as needed.
- Default BabyAI Bot behavior: Alternatively, the bot can replicate the default behavior of the original BabyAI Bot, starting with a subgoal stack that contains only a direct translation of the task description. For instance, the instruction `pick up the red box` is translated into the subgoal stack `[PickupSubgoal, GoNextToSubgoal(red box)]`.

The OmniBot achieves a 100% solve rate across all levels included in the benchmark. This perfect performance provides a robust standard and an effective upper bound for evaluating LLM performance.

## C.6 EXAMPLE OF HOW OMNIBOT'S SUBGOALS STACK CHANGES DURING EXECUTION

We show in the following how the OmniBot can append new subgoals dynamically. We take the example of seed 6 from UnblockPickup environment (see Figure 6) and we start from the default initialization:

- Initial stack: `[DropSubgoal,PickupSubgoal, GoNextToSubgoal(green ball)]`

- Stack at step 6: `[(DropSubgoal), (PickupSubgoal), (GoNextToSubgoal: green ball None), (DropSubgoal), (GoNextToSubgoal: (20,12)), (PickupSubgoal)]`

- Stack at step 7: `[(DropSubgoal), (PickupSubgoal), (GoNextToSubgoal: green ball None), (DropSubgoal), (GoNextToSubgoal: 20, 12))]`

- Stack at step 9: `[(DropSubgoal), (PickupSubgoal), (GoNextToSubgoal: green ball None), (DropSubgoal)]`

- Stack at step 10: `[(DropSubgoal), (PickupSubgoal), (GoNextToSubgoal: green ball None)]`

- Stack at step 15: `[(DropSubgoal), (PickupSubgoal), (GoNextToSubgoal: green ball None), (OpenSubgoal)]`

- Stack at step 16: `[(DropSubgoal), (PickupSubgoal), (GoNextToSubgoal: green ball None)]`

- Stack at step 22: `[(DropSubgoal), (PickupSubgoal)]`

- Final stack after the success of the mission: `[(DropSubgoal)]`

In the example described above, upon completion of the episode, the stack still retains one subgoal that appears to have been unnecessarily added during initialization. This raises a reasonable question: given that the mission objective is solely to pickup the green ball, why include a `DropSubgoal`? This behavior is, in fact, a deliberate safety mechanism implemented to ensure robustness across varied environments and mission configurations. To illustrate this necessity, consider the SynthSeq environment with seed 166, where the OmniBot initially navigates to and picks up a grey ball. While it might seem efficient to save an action to release the ball until absolutely required (such as when needing to pick up another object), immediately dropping the ball prevents mission-critical failures. Without this mechanism, the agent would retain the grey ball, resulting in an absence of the necessary object in the agent's immediate environment, thereby rendering subsequent mission objective "go

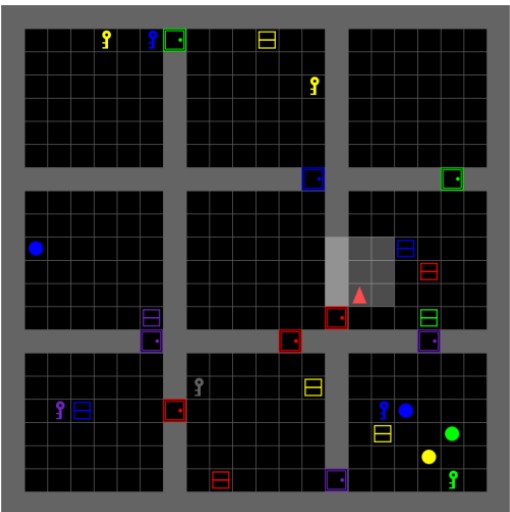

Figure 6: BabyAI UnblockPickup level with seed 8. The mission is to "pickup the green ball". This example provides two type of cases where the OmniBot needs to dynamically add one or many subgoals ot the stack: opening a door and moving an object that prevents from using a door.

to the ball in front of you" impossible to complete. To address this potential failure mode, we have implemented a default behavior that automatically adds a `DropSubgoal` whenever a mission requires picking up an object.

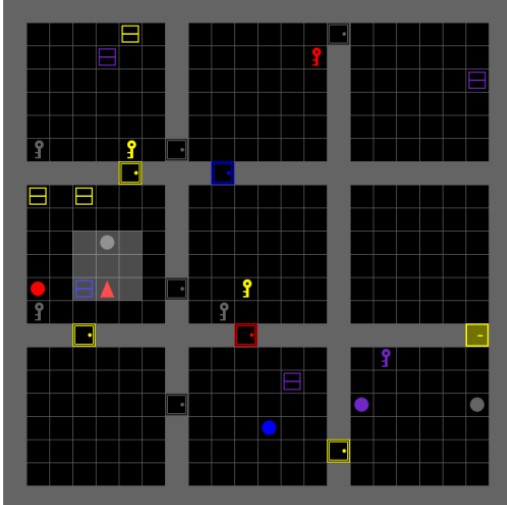

Figure 7: BabyAI SynthSeq level with seed 166. The mission is to "pick up a grey ball and go to the ball in front of you, then go to a box and put a purple box next to the red door ". Even if the mission doesn't ask explicitly to drop the grey ball immediately after picking it up, not doing so will cause the bot to fail the mission since there are no ball in front of the agent anymore.

## C.7 EXAMPLE OF FORMATTING STYLES

We give an example of the result of different formatting styles on the SynthSeq (seed 166, Figure 7).

- **Narrative text**:An agent in a grid world made of 3x3 rooms, each of size 8x8, including the surrounding walls, meaning that effectively, each room is of size 6x6. The total grid size is thus 22x22. Rooms are separated by walls and might contain objects such as keys,

balls, and boxes of different colors. Some walls, connecting two adjacent rooms, have doors. Some doors are unlocked, whereas others need to be unlocked with keys of the same color. The agent can perform 6 actions: left (turn left), right (turn right), forward (move forward), pickup (pickup an object), drop (drop an object), and toggle (open/close a door or a box). Only the forward action changes the agent's position in the grid world. Turning left or right changes the agent's orientation only but not the position. The agent cannot move into a cell that is already occupied by an object, even if the object is one it is trying to interact with. Using a coordinate system where the (0, 0) position is the top-left corner of the grid world, necessarily corresponding to a wall, the coordinates follow the format (x, y), with x denoting the horizontal position in the grid and y denoting the vertical position in the grid, and the agent is initially placed at (4, 12), and is facing, north, the (4, 11) position. There is a yellow box at position (5, 1), an unlocked grey door at position (14, 1), a purple box at position (4, 2), a red key at position (13, 2), a purple box at position (20, 3), a grey key at position (1, 6), a yellow key at position (5, 6), an unlocked grey door at position (7, 6), an unlocked yellow door at position (5, 7), an unlocked blue door at position (9, 7), a yellow box at position (1, 8), a yellow box at position (3, 8), a grey ball at position (4, 10), a red ball at position (1, 12), a blue box at position (3, 12), an unlocked grey door at position (7, 12), a yellow key at position (10, 12), a grey key at position (1, 13), a grey key at position (9, 13), an unlocked yellow door at position (3, 14), an unlocked red door at position (10, 14), a locked yellow door at position (20, 14), a purple key at position (16, 15), a purple box at position (12, 16), an unlocked grey door at position (7, 17), a purple ball at position (15, 17), a grey ball at position (20, 17), a blue ball at position (11, 18), and an unlocked yellow door at position (14, 19). The agent's mission is 'pick up a grey ball and go to the ball in front of you, then go to a box and put a purple box next to the red door.

- **Structured text**: An agent is in a grid world consisting of one or more rooms. All rooms in the same grid world are squares of identical size and are organized in a square grid layout. Rooms are separated by walls and might contain objects such as keys, balls, and boxes of different colors. Some walls, connecting two adjacent rooms, have doors. Some doors are unlocked, whereas others need to be unlocked with keys of the same color. The agent can perform 6 actions:

    - left (turn left),
    - right (turn right),
    - forward (move forward),
    - pickup (pickup an object),
    - drop (drop an object),
    - toggle (open/close a door or a box).

  Only the forward action changes the agent's position in the grid world. Turning left or right changes the agent's orientation only but not the position. The agent cannot move into a cell that is already occupied by an object, even if the object is one it is trying to interact with. Using a coordinate system where the (0, 0) position is the top-left corner of the grid world, necessarily corresponding to a wall, the coordinates follow the format (x, y), with x denoting the horizontal position in the grid and y denoting the vertical position in the grid.

  These are the specifics regarding this environment:

    - Number of rooms: 3x3
    - Size of each room (including walls): 8x8
    - Effective room size (excluding walls): 6x6
    - Total grid size: 22x22
    - Agent initial position: (4, 12)
    - Agent facing direction: north (toward (4, 11))
    - Objects in environment:
        * box, color=yellow, position=(5, 1)
        * door, color=grey, position=(14, 1), locked=False
        * box, color=purple, position=(4, 2)
        * key, color=red, position=(13, 2)
        * box, color=purple, position=(20, 3)

           * key, color=grey, position=(1, 6)
           * key, color=yellow, position=(5, 6)
           * door, color=grey, position=(7, 6), locked=False
           * door, color=yellow, position=(5, 7), locked=False
           * door, color=blue, position=(9, 7), locked=False
           * box, color=yellow, position=(1, 8)
           * box, color=yellow, position=(3, 8)
           * ball, color=grey, position=(4, 10)
           * ball, color=red, position=(1, 12)
           * box, color=blue, position=(3, 12)
           * door, color=grey, position=(7, 12), locked=False
           * key, color=yellow, position=(10, 12)
           * key, color=grey, position=(1, 13)
           * key, color=grey, position=(9, 13)
           * door, color=yellow, position=(3, 14), locked=False
           * door, color=red, position=(10, 14), locked=False
           * door, color=yellow, position=(20, 14), locked=True
           * key, color=purple, position=(16, 15)
           * box, color=purple, position=(12, 16)
           * door, color=grey, position=(7, 17), locked=False
           * ball, color=purple, position=(15, 17)
           * ball, color=grey, position=(20, 17)
           * ball, color=blue, position=(11, 18)
           * door, color=yellow, position=(14, 19), locked=False

      – Mission: 'pick up a grey ball and go to the ball in front of you, then go to a box and put a purple box next to the red door.'

- **JSON**:

```
{
  "context": "An agent is in a grid world consisting of one or
  ↪  more rooms. All rooms in the same grid world are squares of
  ↪  identical size and are organized in a square grid layout.
  ↪  Rooms are separated by walls and might contain objects such
  ↪  as keys, balls, and boxes of different colors. Some walls,
  ↪  connecting two adjacent rooms, have doors. Some doors are
  ↪  unlocked, whereas others need to be unlocked with keys of
  ↪  the same color. The agent can perform 6 actions: left (turn
  ↪  left), right (turn right), forward (move forward), pickup
  ↪  (pickup an object), drop (drop an object), and toggle
  ↪  (open/close a door or a box). Only the forward action changes
  ↪  the agent's position in the grid world. Turning left or
  ↪  right changes the agent's orientation only but not the
  ↪  position. The agent cannot move into a cell that is already
  ↪  occupied by an object, even if the object is one it is
  ↪  trying to interact with. Using a coordinate system where the
  ↪  (0, 0) position is the top-left corner of the grid world,
  ↪  necessarily corresponding to a wall, the coordinates follow
  ↪  the format (x, y), with x denoting the horizontal position
  ↪  in the grid and y denoting the vertical position in the
  ↪  grid,\n\nThese are the specifics regarding this environment:
  ↪  \n\n",
  "config": {
    "num_rooms": [3, 3],
    "room_size_incl_walls": [8, 8],
    "room_size_excl_walls": [6, 6],
    "grid_size": [22, 22],
    "agent_initial_pos": [4, 12],
    "agent_front_pos": [4, 11],
    "agent_direction": {
      "index": 3,
```

```
1242              "name": "north"
1243            },
1244            "objects": [
1245              {"type": "box", "color": "yellow", "position": [5, 1]},
1246              {"type": "door", "color": "grey", "position": [14, 1],
1247            ↪   "locked": false},
1248              {"type": "box", "color": "purple", "position": [4, 2]},
1249              {"type": "key", "color": "red", "position": [13, 2]},
1250              {"type": "box", "color": "purple", "position": [20, 3]},
1251              {"type": "key", "color": "grey", "position": [1, 6]},
1252              {"type": "key", "color": "yellow", "position": [5, 6]},
1253              {"type": "door", "color": "grey", "position": [7, 6],
1254            ↪   "locked": false},
1255              {"type": "door", "color": "yellow", "position": [5, 7],
1256            ↪   "locked": false},
1257              {"type": "door", "color": "blue", "position": [9, 7],
1258            ↪   "locked": false},
1259              {"type": "box", "color": "yellow", "position": [1, 8]},
1260              {"type": "box", "color": "yellow", "position": [3, 8]},
1261              {"type": "ball", "color": "grey", "position": [4, 10]},
1262              {"type": "ball", "color": "red", "position": [1, 12]},
1263              {"type": "box", "color": "blue", "position": [3, 12]},
1264              {"type": "door", "color": "grey", "position": [7, 12],
1265            ↪   "locked": false},
1266              {"type": "key", "color": "yellow", "position": [10, 12]},
1267              {"type": "key", "color": "grey", "position": [1, 13]},
1268              {"type": "key", "color": "grey", "position": [9, 13]},
1269              {"type": "door", "color": "yellow", "position": [3, 14],
1270            ↪   "locked": false},
1271              {"type": "door", "color": "red", "position": [10, 14],
1272            ↪   "locked": false},
1273              {"type": "door", "color": "yellow", "position": [20, 14],
1274            ↪   "locked": true},
1275              {"type": "key", "color": "purple", "position": [16, 15]},
1276              {"type": "box", "color": "purple", "position": [12, 16]},
1277              {"type": "door", "color": "grey", "position": [7, 17],
1278            ↪   "locked": false},
1279              {"type": "ball", "color": "purple", "position": [15, 17]},
1280              {"type": "ball", "color": "grey", "position": [20, 17]},
1281              {"type": "ball", "color": "blue", "position": [11, 18]},
1282              {"type": "door", "color": "yellow", "position": [14, 19],
1283            ↪   "locked": false}
1284            ],
1285            "mission": "pick up a grey ball and go to the ball in front of
1286          ↪  you, then go to a box and put a purple box next to the red
1287          ↪  door"
1288          }
1289        }
```

### C.7.1 IMPACT OF FORMATTING ON THE MODELS

We compare the models' performance on the Predict task varying the formatting, but fixing the prompter to be ToT.

The comparison averaged over all models is given in the Table 6. It shows that the structured text formatting works the best in the given settings.

Table 6: Comparison of Formatting styles based on Success Rate and L1 Distance (lower is better)

| Formatting Style | Success Rate (%) | Manhattan Distance (↓) |
|---|---|---|
| Narrative | 56.78 | 2.34 |
| Json | 59.28 | 2.65 |
| Structured text | **61.08** | **2.24** |

# D  DETAILS ON FPI EXPERIMENTS

To complement our claim, we introduce FPI (Full mission, Partial observability, Interactive) experiments that test LLMs under more realistic embodied agent conditions. Unlike in PPD, where models receive complete environment information upfront, the FPI setting introduces three key constraints:

- **Full mission completion:** Models must solve complete missions from start to finish, rather than individual subtasks, requiring sustained reasoning over longer horizons.
- **Partial observability:** Models receive only limited environmental information based on the agent's current position and field of view, mimicking the partial observability constraints of the original BabyAI platform. This requires models to maintain internal representations of unexplored areas and reason about uncertainty.
- **Interactive execution:** Models must generate actions step-by-step while receiving environmental feedback, allowing for dynamic replanning but requiring real-time decision making under uncertainty.

The FPI experiments use the same 16 BabyAI levels, but evaluate models across four difficulty categories with varying complexity parameters. For each difficulty level, we test configurations with different agent view sizes (indicated by the numbers 3, 5, 7, 9 in Table 3).

Table 3 reveals several key findings from the FPI evaluation. Both Claude-Sonnet-4 and GPT-5 demonstrate strong performance on Easy tasks, achieving success rates between 59-74%, with Claude-Sonnet-4 showing a slight advantage in the simpler configurations. However, performance degrades substantially as task complexity increases. On Moderate difficulty tasks, both models experience a significant drop to approximately 30% success rate, while maintaining reasonable efficiency levels around 27-43%.

The transition to Hard difficulty tasks shows mixed results: while success rates remain in the 36-43% range, there's notable variability between different complexity configurations. Interestingly, both models show comparable performance across Hard task variants, suggesting that the partial observability constraints level the playing field between different model architectures.

The most striking finding emerges in Very Hard tasks, where both models struggle significantly, achieving only 10-12% success rates. This represents a dramatic performance cliff compared to Hard tasks, indicating that the combination of long-horizon planning, partial observability, and interactive execution creates a substantial challenge for current LLMs. Despite the low success rates, GPT-5 maintains slightly better efficiency (7-9%) compared to Claude-Sonnet-4 (4%) on the most challenging configurations, suggesting different failure modes between the models.

# E  COMPARISON OF PROMPTERS

We began by evaluating the prompting strategies described in Section 4.1 to determine which approach performs best on our tasks. Specifically, we focused on the Predict task, using success rate and L1 distance as our evaluation metrics. To this end, we compared all prompting strategies on the GoToObj and BossLevel missions across 20 random seeds.

As shown in Table 7, when averaged over all models, the Tree-of-Thought (ToT) prompting strategy consistently outperformed the others, achieving higher success rates and lower L1 distances. This result is intuitive, as the ToT method encourages the model to explore multiple reasoning paths in a structured, branching manner, which is an advantage when tackling embodied reasoning tasks.

Given its strong performance, we adopt the ToT prompting strategy for all subsequent experiments.

# F  EVALUATION METRIC DETAILS

## F.1  DETAILED SUMMARY

This section provides a detailed summary of the metrics used to evaluate LLM performance across the three BABYBENCH tasks: Predict, Plan, and Decompose. Table 8 lists each metric, provides a

Table 7: Comparison of Prompters based on Success Rate and L1 Distance (lower is better)

| Prompter | Success Rate (%) | Manhattan Distance ($\downarrow$) |
|---|---|---|
| Zero-Shot | 59.29 | 3.09 |
| Few-Shot | 59.05 | 2.47 |
| CoT | 61.07 | 2.79 |
| ToT | **62.62** | **2.41** |

description of how it is calculated, and specifies the particular reasoning ability or output quality it is designed to measure. A more extensive discussion of the rationale behind these metrics and the evaluation process is available in the remainder of this section.

Table 8: Details of Evaluation Metrics for BABYBENCH tasks

| Task | Metric | Description | What it Measures |
|---|---|---|---|
| **Predict** | Success Rate | Proportion of predictions where the agent's final state description exactly matches the correct target state description. | Exact prediction accuracy of the final state. |
| | Manhattan Distance | The L1 distance between the predicted agent position and the correct target agent position, calculated for incorrect predictions. | Degree of spatial error in predicting the final agent location. |
| **Plan** | Success Rate | Proportion of LLM-generated action sequences that successfully achieve the target subgoal when executed in the environment. | Task completion rate for low-level planning. |
| | Efficiency Ratio | For successful plans, the ratio of the number of actions in the OmniBot's optimal sequence to the number of actions in the LLM's sequence. | Optimality and efficiency of the generated action plan (high efficiency is desired). |
| **Decompose** | Comprehension Rate | Proportion of tasks where the final mission goal is achieved when the OmniBot executes the LLM's subgoal sequence, allowing the bot to add necessary subgoals. | Understanding of the overall mission objective (maximal assistance allowed). |
| | Precision Rate | Proportion of tasks where the final mission goal is achieved when the OmniBot executes *only* the LLM's subgoal sequence, adding no subgoals. | Completeness and correctness of the decomposition (zero assistance allowed). |
| | Assistance Curve Integral (ACI) | The area under the curve plotting task success rate against the maximum number of additional subgoals ($k$) the OmniBot is allowed to add (from $k = 0$ up to unlimited assistance). | Performance across varying levels of required external subgoal assistance. |

## F.2 PREDICT EVALUATION

Let $(x_i, y_i)_{1 \leq i \leq n}$ the positions predicted by the LLM and $(\hat{x}_i, \hat{y}_i)_{1 \leq i \leq n}$ the real positions after the sequences of actions given in the prompts respectively. Then:

**Success Rate:**

$$\frac{1}{n} \sum_{i=1}^{n} \mathbf{1} \left( x_i = \hat{x}_i \ \wedge \ y_i = \hat{y}_i \right) \tag{1}$$

**Manhattan Distance:**

$$\frac{1}{n} \sum_{i=1}^{n} \left( |x_i - \hat{x}_i| + |y_i - \hat{y}_i| \right) \tag{2}$$

## F.3 PLAN EVALUATION

Let $(\mathbf{a}_i)_{1 \leq i \leq n}$ the sequence of actions and $(\hat{x}_i, \hat{y}_i)_{1 \leq i \leq n}$ the ground truth position to achieve respectively. Then:

$$\text{SuccessRate} = \begin{cases} 1, & \text{if the execution of } (\mathbf{a}_i)_{1 \leq i \leq n} \text{ reaches the goal state } (\hat{x}_n, \hat{y}_n), \\ 0, & \text{otherwise.} \end{cases}$$

## F.4 DECOMPOSITION EVALUATION

In this section, we represent subgoal sequences using a stack notation, where the most prioritized subgoal appears at the top of the stack (i.e., last in the list). Additionnaly, while in practice the LLM are expected to output subgoals in the format `GoNextToSubgoal((x,y))`, the following discussion will use descriptions `GoNextToSubgoal(grey ball)` instead of coordinates to ease the understanding of the examples given.

### F.4.1 WHY NOT COMPARE OMNIBOT AND LLM DECOMPOSITIONS DIRECTLY?

A natural idea might be to evaluate the LLM's decomposition quality by comparing its proposed subgoal stack to that of the OmniBot. However, this comparison is not appropriate due to fundamental differences in how subgoals are generated and updated by each system.

The OmniBot interacts with the environment in real time and can dynamically update its subgoal stack as it gathers new information or encounters obstacles. In contrast, the LLM generates a fixed list of subgoals in advance, without the ability to revise them during execution. Since the LLM has full visibility of the grid at generation time, it is theoretically possible to anticipate all necessary steps and contingencies upfront.

Even assuming a perfect LLM with ideal reasoning capabilities, its generated subgoals will generally not match those of the OmniBot. This is not a flaw in the LLM's reasoning, but rather a reflection of the static nature of its output compared to the OmniBot's dynamic, adaptive behavior.

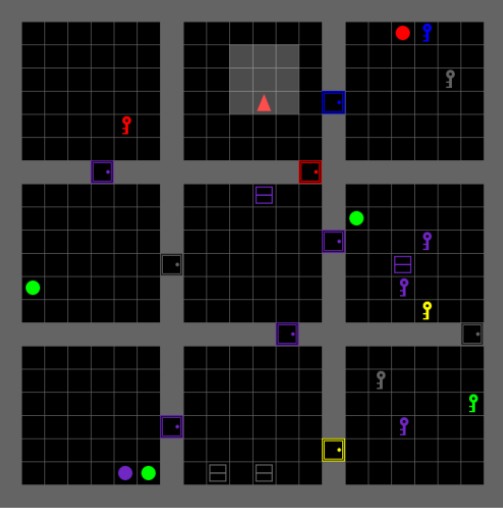

Figure 8: BabyAI GoTo level with seed 54. The task is to "go to the blue key". This example illustrates why the OmniBot and a perfect LLM may produce different subgoal sequences.

Consider the example shown in Figure 8, where the agent must go to the blue key. The OmniBot begins the episode with the subgoal stack:

```
[GoNextToSubgoal(blue key)]
```

As the episode progresses, the OmniBot moves toward the blue key. Upon reaching the blue door blocking the path, it appends a new subgoal:

```
[GoNextToSubgoal(blue key), OpenSubgoal]
```

After opening the door, the OmniBot removes the now-completed `OpenSubgoal`, returning the stack to its original form before completing the task.

In contrast, the LLM must anticipate the need to open the door and generate the complete subgoal sequence upfront:

```
[GoNextToSubgoal(blue key), OpenSubgoal, GoNextToSubgoal(blue door)]
```

This discrepancy illustrates the fundamental mismatch in subgoal structures: the OmniBot leverages a dynamic subgoal policy, while the LLM must commit to a static plan.

Therefore, a direct comparison between the LLM's subgoals and those of the OmniBot is misleading for two key reasons:

1. The OmniBot's subgoal stack is dynamic and evolves over time.

2. A perfect LLM will include subgoals that are never explicitly added by the OmniBot, since they are only needed transiently and are handled reactively by the bot.

Hence, directly matching subgoal stacks would penalize even optimal LLM predictions, making such a metric unsuitable for evaluating decomposition quality.

### F.4.2 THE EVALUATION PROCESS

As direct comparison between subgoals generated by the LLM and those used by the OmniBot is not meaningful (as discussed in the previous section), we propose a new evaluation strategy tailored specifically to the BABYBENCH Decompose task. This procedure involves the following steps:

1. We prompt the LLM to decompose the level's instruction into a sequence of subgoals.

2. The bot is initialized with this sequence of subgoals.

3. The bot then executes an episode. During execution, we record the number of additional subgoals the bot needs to append in order to complete the task (whether by success, reaching the maximum number of steps, or depleting the subgoal stack).

4. For each episode, we record the success status and the number of subgoals added. These values are used to compute three metrics: Comprehension Rate (CR), Precision Rate (PR), and Assistance Curve Integral (ACI).

### F.4.3 WHAT DOES COMPREHENSION RATE REALLY MEASURE?

The Comprehension Rate (CR) measures the proportion of episodes successfully completed, regardless of how many subgoals the OmniBot needs to add during execution. To achieve a high CR, it suffices for the LLM to generate a plausible decomposition of the instruction into subgoals. As noted in Section 3.3, such a decomposition always enables the OmniBot to solve the level successfully, even if the initial stack does not account for the specific obstacles present in the environment. For example, in Figure 8, the LLM simply needs to understand that the level's mission `go to the blue key` can be reformulated as the subgoal `GoNextTo(blue key)` to positively contribute to the CR metric.

This makes CR a relatively undemanding metric: it allows the OmniBot to freely intervene and compensate for incomplete or imprecise LLM outputs, and it does not require the LLM to reason about the environment's layout. Rather, the LLM only needs to reformulate the instruction mechanically into a sequence of subgoals from a predefined set. For this reason, we interpret CR as a measure of the LLM's task comprehension: its ability to understand and translate a high-level instruction into a structured goal representation.

### F.4.4 WHAT DOES PRECISION RATE REALLY MEASURE?

The *Precision Rate* (PR) quantifies the proportion of episodes successfully completed without the OmniBot needing to insert any additional subgoals during execution. A high PR indicates that the language model has fully anticipated all necessary steps and obstacles, explicitly encoding them as subgoals in its initial output.

For instance, in Figure 8, the shortest correct decomposition is `[GoNextTo(blue key),` `OpenSubgoal, GoNextTo(blue door)]`, which the model must generate unaided to count toward PR.

PR is especially challenging in complex scenarios that involve multiple intermediate steps. Consider the example in Figure 9: with OmniBot assistance, the following short sequence leads to succeeding the mission:

```
[DropSubgoal, GoNextToSubgoal(red box), PickupSubgoal,
 GoNextToSubgoal(yellow ball), OpenSubgoal, GoNextToSubgoal(green
                           door)].
```

However, completing the task without any subgoal insertion by the bot requires the model to generate a significantly longer and more detailed sequence:

```
[DropSubgoal, GoNextToSubgoal(red box), PickUp,
 GoNextToSubgoal(yellow ball), OpenSubgoal, GoNextToSubgoal(blue
     door), PickupSubgoal, GoNextTo(blue key), DropSubgoal,
 GoNextTo(drop position), OpenSubgoal, GoNextToSubgoal(green door),
     OpenSubgoal, GoNextToSubgoal(purple door), OpenSubgoal,
 GoNextToSubgoal(blue door), PickupSubgoal, GoNextToSubgoal(yellow
   ball), OpenSubgoal, GoNextToSubgoal(green door), OpenSubgoal,
               GoNextToSubgoal(grey door)].
```

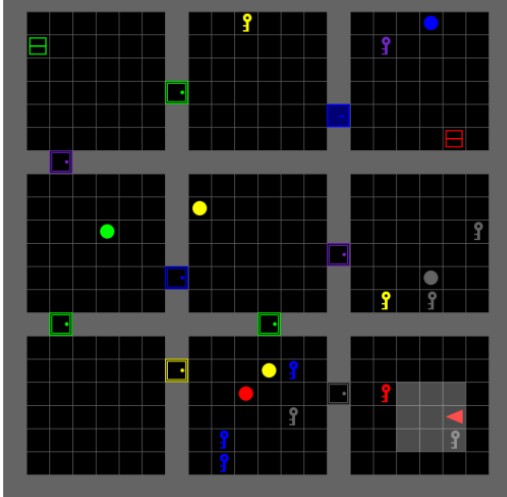

Figure 9: BabyAI BossLevel (seed 81). The task is to "open a green door, then put a yellow ball next to the red box". This mission illustrates a clear contrast in planning complexity: with unlimited assistance from the OmniBot, a concise list of just 5 subgoals is sufficient for success, whereas achieving the same outcome without any assistance requires an extended sequence of 22 subgoals.

### F.4.5 WHAT DOES ASSISTANCE CURVE INTEGRAL REALLY MEASURE?

While the *Comprehension Rate* measures the success rate with unlimited help from the OmniBot, and the *Precision Rate* measures success without any assistance, the *Assistance Curve Integral* (ACI) provides a smooth interpolation between these two extremes. Rather than selecting an arbitrary

threshold $k$, ACI integrates the success rate as a function of the number of subgoals added, from $0$ to a predefined *limit*. This limit corresponds to the number of subgoals the OmniBot would typically insert under its default initialization (Section 3.3). The integral is then normalized by dividing by *limit*, such that a perfect model achieves an ACI of 1.

A high ACI indicates that the model typically requires little assistance to solve tasks, whereas a low ACI suggests heavy reliance on OmniBot interventions.

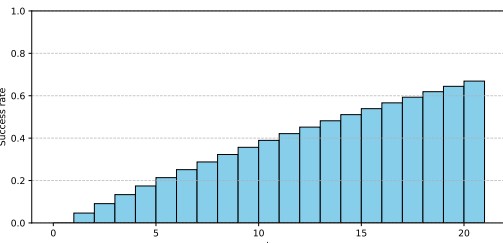

Figure 10: Example bar chart of the success rate as a function of the number of subgoals added by the OmniBot. The blue area represents the unnormalized ACI.

## G PREDICT TASK ANALYSIS

Table 9 presents the performance of each model across a set of targeted competencies. These results reinforce the observation that Claude demonstrates a deeper understanding of how individual actions influence both the agent and the environment. At a more granular level, the table highlights specific weaknesses of other models. For instance, while models such as DeepSeek-R1-Distilled, GPT-5, and Qwen3 perform well in single-room environments, their accuracy drops substantially in maze-like scenarios. Furthermore, for all models, the success rates on *Open*, *Pickup*, and *Put* instructions are comparable to those on *Go To* instructions. This is expected, as these actions are typically executed by first completing a *GoNextTo* subgoal, followed by the respective action; since they do not involve a change in position, performance largely depends on the model's ability to predict the outcomes of navigation primitives such as *forward* and *left/right*. Finally, with the exception of Claude, all models exhibit significant difficulties when processing sequences of instructions, which often require longer navigation paths and interaction with multiple objects.

In Appendix I, we can see the difference between the reasonings of Claude and DeepSeek-R1-Distilled model on a very hard level. While Anthropic's model solely focus on calculating the next state at each step as a human would do for the same task, R1-Distilled model starts deviating from the main task, sometimes even questioning the exactitude of the description of the grid given in the prompt.

Table 9: Model Performance on Specific BabyAI Competencies (Success Rate %)

| Competency | Claude | DeepSeek-R1-Distill | GPT-5 | Meta-Llama-3.1-405B | Meta-Llama-3.1-70B | Meta-Llama-3.1-8B | Qwen |
|---|---|---|---|---|---|---|---|
| Room Navigation | 99.20 | 89.40 | 95.70 | 98.30 | 55.50 | 6.00 | 88.30 |
| Maze Navigation | 84.50 | 50.30 | 81.00 | 71.00 | 45.30 | 3.20 | 50.80 |
| Unblocking | 78.00 | 42.90 | 71.60 | 68.00 | 49.40 | 1.30 | 45.70 |
| Unlocking | 77.50 | 41.50 | 73.80 | 66.20 | 50.00 | 1.80 | 44.40 |
| Go To Instructions | 88.70 | 54.20 | 80.60 | 70.00 | 50.70 | 5.00 | 55.40 |
| Open Instructions | 81.00 | 50.60 | 79.50 | 70.00 | 49.40 | 5.50 | 52.20 |
| Pickup Instructions | 80.70 | 51.70 | 78.60 | 75.00 | 49.10 | 2.70 | 55.40 |
| Put Instructions | 81.00 | 48.10 | 74.70 | 73.00 | 46.20 | 1.30 | 49.40 |
| Location Language | 88.60 | 58.30 | 82.00 | 77.10 | 48.20 | 4.60 | 59.20 |
| Sequences of Commands | 76.70 | 29.20 | 66.70 | 56.70 | 40.90 | 0.00 | 31.50 |

## G.1 ARITHMETIC RESOLUTION OF PREDICT

Succesful LLMs start from an initial state $(x_0, y_0, \theta_0)$ with $\theta \in \{0, 1, 2, 3\}$ denoting orientation, and then apply simple updates:

$$(x_{t+1}, y_{t+1}, \theta_{t+1}) = \begin{cases} (x_t + \mathbf{1}_{\theta_t=0} - \mathbf{1}_{\theta_t=2},\ y_t + \mathbf{1}_{\theta_t=3} - \mathbf{1}_{\theta_t=1},\ \theta_t), & \text{if action} = \texttt{forward}, \\ (x_t, y_t, (\theta_t + 1) \bmod 4), & \text{if action} = \texttt{right}, \\ (x_t, y_t, (\theta_t - 1) \bmod 4), & \text{if action} = \texttt{left}, \\ (x_t, y_t, \theta_t), & \text{otherwise.} \end{cases}$$

# H PLAN TASK ANALYSIS

## H.1 OVERALL EVALUATION

Figure 11(a) shows that the highest success rate in the path planning task is achieved by reasoning-capable models such as GPT-5, DeepSeek Distilled model, Claude, and Qwen3. These models likely benefit from their ability to handle multi-step logical dependencies, enabling them to generate plans that lead to the correct goal more consistently.

In contrast, Figure 11(b) reveals an interesting observation: even models with relatively low success rates, such as Llama-405B and Llama-70B , can still produce efficient action sequences when they do succeed. This suggests that while these models may struggle with reasoning over complex environments to consistently generate correct plans, their output - when correct - tends to be concise and optimized in terms of action length or redundancy.

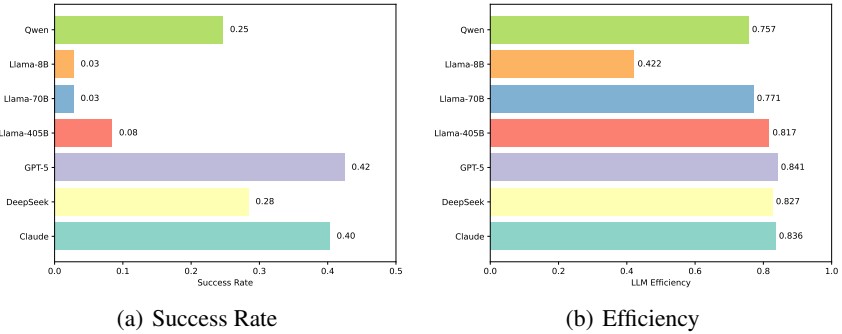

(a) Success Rate        (b) Efficiency

Figure 11: Comparison of LLM models based on success rate and efficiency.

## H.2 IMPACT OF OBSTACLES ON PLANNING TASK

Figure 12 shows that the main factor determining the failure of the models on our Plan task is the size of the grid. Within the same size the number of obstacles does not affect the success rate of the model, since there is no trend for the models to have a higher success rate within the same grid size.

# I EXAMPLES OF LLM REASONING TRACES

This appendix provides illustrative examples of the reasoning processes generated by different LLMs when tackling BABYBENCH tasks, as discussed in Section 4. These traces offer qualitative insights into model behavior, complementing the quantitative results presented in the main text and highlighting differences in problem-solving approaches.

## I.1 PREDICT: BOSSLEVEL (SEED 47)

The mission is described in Figure 13.

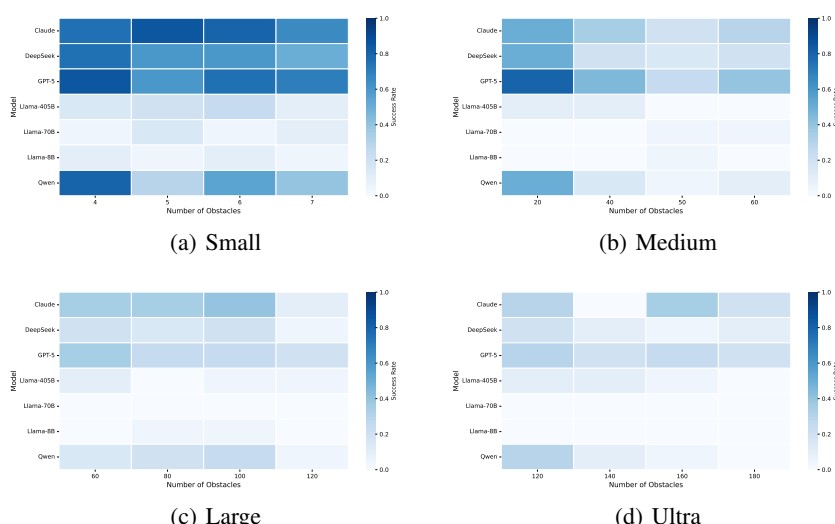

(a) Small          (b) Medium

(c) Large          (d) Ultra

Figure 12: CR heatmaps per difficulty level, grouped by model (rows) and number of obstacles (columns).

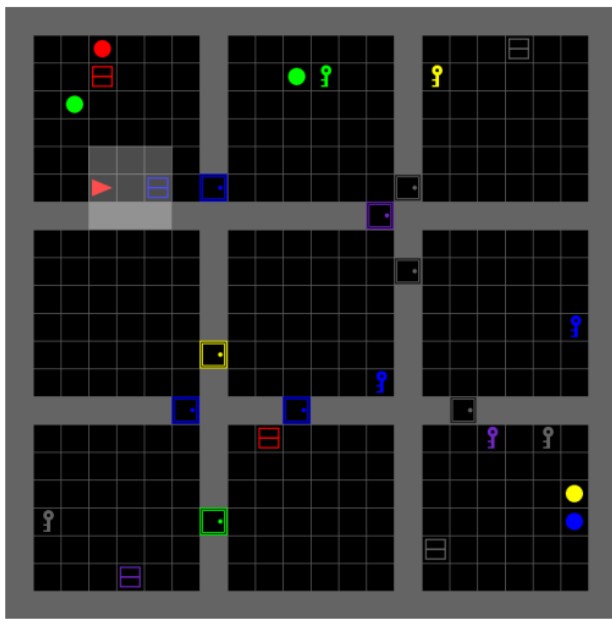

Figure 13: BossLevel (seed 47). The mission is to pickup the blue ball. R1 fails this level in the predict task while Claude succeed.

### I.1.1 CLAUDE 4 SONNET PREDICT TRACE ✓

Claude demonstrates systematic, methodical tracking of each action without uncertainty:

```
Initial State Analysis
- Agent's initial position: (3, 6)
- Agent's initial direction: east (toward (4, 6))

# Action Sequence Analysis

## Initial state: (3, 6) facing east
```

```
1. **forward**
   - Moving east from (3, 6) would take the agent to (4, 6)
   - Checking for obstacles at (4, 6): None mentioned
   - New state: (4, 6) facing east

2. **left**
   - From east, turning left would make the agent face north
   - New state: (4, 6) facing north

[... Actions 3-9 tracked with same systematic approach ...]

10. **toggle**
    - The agent is at (6, 6) facing east
    - Looking at position (7, 6), which has a blue door (unlocked)
    - The agent toggles the door, presumably opening it
    - New state: (6, 6) facing east, blue door at (7, 6) is now open

11. **forward**
    - Moving east from (6, 6) would take the agent to (7, 6)
    - The door at (7, 6) is now open, so the agent can move through
    - New state: (7, 6) facing east

12-17. **forward** (6 times)
    - Moving east from (7, 6) through (8, 6), (9, 6), (10, 6),
      (11, 6), (12, 6), to (13, 6)
    - No obstacles mentioned in this path
    - New state: (13, 6) facing east

[... Continues with same methodical approach ...]

33. **toggle**
    - The agent is at (16, 13) facing south
    - Looking at position (16, 14), which has a grey door (unlocked)
    - The agent toggles the door, presumably opening it
    - New state: (16, 13) facing south, grey door at (16, 14) is now
    ↪  open

[...]

43. **pickup**
    - The agent is at (19, 18) facing east
    - Looking at position (20, 18), which has a blue ball
    - The agent picks up the blue ball
    - New state: (19, 18) facing east, holding blue ball

The agent's final state is: ((19, 18), 0)
```

Claude's trace shows: (1) consistent state tracking without confusion, (2) explicit obstacle checking at each step, (3) proper handling of door interactions, (4) efficient grouping of repeated actions, and (5) confident final answer without hedging or uncertainty.

### I.1.2   DEEPSEEK-R1-DISTILLED PREDICT TRACE ✗

DeepSeek-R1-Distilled demonstrates confusion about spatial reasoning and frequent self-doubt:

```
<think>
Alright, let's try to figure out where the agent ends up after
executing the given sequence of actions. The mission is to pick up
the blue ball.

First, I'll start by noting the initial state. The agent begins at
position (3, 6) facing east. So, direction 0.

[... Actions 1-5 tracked correctly ...]
```

```
6. **Action: forward**
   Moves to (6,5). Now, this position is a door at (7,6) but
   wait, that's east. Hmm, maybe I'm getting the door position
   wrong. Wait, (7,6) is a door, but the agent is at (6,5).
   Moving forward from (6,5) east would take it to (7,5).
   Wait, actually, each room is 8x8, so perhaps moving beyond
   the current room.

Wait, the grid is 22x22, and each room is 8x8, but they're
arranged in 3x3, so coordinates go up to around 24? Maybe I'm
miscalculating. Let me think again.

[...]

Wait, perhaps I should map each room's coordinates. The agent
starts in the center room, which spans from (3,3) to (10,10), maybe?
Or perhaps each room is 8x8, so rooms are at (0-7, 0-7),
(8-15, 0-7), etc. But I'm getting a bit confused here.
Maybe I should just track each movement step by step without
worrying about the rooms too much.

[... Continues tracking actions with uncertainty ...]

24. **Action: left**
    Facing south, turning left makes the agent face east (direction 0).
    Wait, no. Facing south, left turn would face east towards
    south? Wait, let me think. [...]
    Wait, no. When you turn left from facing south, you turn
    towards east, yes. So direction becomes east (0).

[...]

30. **Action: forward**
    To (22,6). But the grid is 22x22, so (22,6) is at the edge.
    Maybe the agent can't move beyond that. Wait, the grid is
    22x22, with coordinates from (0,0) to (21,21). So (22,6) is outside.
    Therefore, the agent can't move forward here. So the agent
    stays at (21,6).

    Wait, but I'm not sure. Maybe each room is 8x8, and the
    agent is moving through multiple rooms. But perhaps I'm
    overcomplicating.

[...]

Wait, perhaps I'm getting lost. Let me recount the actions.

Wait, the action sequence given is 41 actions, and I might
have miscounted earlier.

Let me try a different approach, perhaps making a table.

But given time constraints, I'll summarize:

[... Gives up on systematic tracking ...]

Thus, the final state is at (20,18), facing whatever direction after
the last movement. [...]

I think the final direction is south, so direction 1, and
position (20,18). But I'm not entirely certain without
tracking each step precisely.
</think>

The agent's final state is: ((20, 18), 1)
```

The trace reveals constant self-doubt ("Wait", "I'm not sure", "Maybe I'm miscalculating"), confusion about basic environment structure, and ultimately abandoning systematic tracking in favor of guessing the final state. Despite the uncertainty, the model coincidentally arrives at a correct answer.

## I.2 DECOMPOSE: SYNTHSEQ (SEED 166)

The mission is described in Figure 7

### I.2.1 CLAUDE 4 SONNET DECOMPOSE TRACE ✗

Claude demonstrates a structured approach but fails due to incorrect coordinate extraction:

```
Step-by-Step Planning for MiniGrid Task

## Step 1: Identify the mission goal
The mission is: "go to the purple box after you open a grey door"
This means we need to:
1. Find and open a grey door
2. After that, go to the purple box

## Step 2: Identify relevant objects and obstacles
Relevant objects:
- Grey doors: at (7, 11) [unlocked] and (7, 19) [unlocked]
- Purple box: at (17, 3)
- Agent starts at: (8, 18) facing north

## Step 3: Determine the necessary interactions
1. We need to open one of the grey doors
2. After opening a grey door, we need to go to the purple box at (17, 3)

## Step 4: Plan at least two options of movement
Option 1: Go to the nearby grey door at (7, 19) [...]
Option 2: Go to the other grey door at (7, 11) [...]

## Step 5: Choose the best option
Option 1 is better because:
- The grey door at (7, 19) is closer to the agent's starting position
- Both doors are unlocked, so there's no advantage [...]

## Step 6 & 7: Convert and sequence the subgoals
Let's break this down into specific subgoals:
1. First, we need to go next to the grey door at (7, 19)
   - Subgoal: (GoNextToSubgoal, (7, 19))
2. Next, we need to open this grey door
   - Subgoal: (OpenSubgoal)
3. Finally, we need to go to the purple box at (17, 3)
   - Subgoal: (GoNextToSubgoal, (17, 3))

<START>
(GoNextToSubgoal, (7, 19))
(OpenSubgoal)
(GoNextToSubgoal, (17, 3))
<END>
```

Despite the systematic approach and correct reasoning about door selection, Claude fails by incorrectly identifying the purple box location as (17, 3) when it's actually at (5, 16), leading to an incorrect final subgoal. This demonstrates how structured reasoning can still fail due to simple perceptual errors.

### I.2.2 QWEN3 DECOMPOSE TRACE ▲

Despite ultimately producing a correct decomposition, Qwen3's reasoning shows excessive uncertainty and redundant exploration:

```
<think>
Okay, let's see. The mission is "go to the purple box after you open
a grey door." So the agent needs to first open a grey door and then
reach the purple box.

First, I need to figure out where the purple box is and where the
grey door is. From the objects list, the purple box is at (5,16).
The grey door is at (7,11) and it's unlocked. So the agent needs to
open that door first. Wait, but the grey door is already unlocked?
Wait, the description says "door, color=grey, position=(7, 11),
locked=False". So it's open or closed? Wait, in the environment,
if a door is unlocked but closed, you need to toggle to open it.
[...]

So first, to reach the grey door at (7,11), the agent has to get there.
[...] The agent starts at (8,18) facing north. The grid is 22x22.
The rooms are arranged in a 3x3 grid, each 8x8. Let's try to map
out the rooms. Maybe the agent is in the middle room? Not sure.
[...]

But maybe the best approach is to plan the path step by step.
[...] However, there might be walls in the rooms. Since the rooms
are 8x8 squares, including walls, the walls would be at the edges.
[...]

Wait, the grey door is at (7,11). Let me see if there's a door
connecting the agent's starting room to the room where the grey
door is located. [...]

Alternatively, perhaps the agent can directly reach (7,11) by
going west and north. [...]

This is getting complicated. Maybe the agent can first move to
the green door at (9,7), then through it to the adjacent room. [...]

Wait, no, doors are either open or closed. If the door is open,
the agent can pass through without toggling. [...]

This is getting too detailed. Maybe the agent needs to first
go to (9,7), then from there to (7,11). Let's plan the steps:
[...]

Alternatively, maybe there's a direct path from (8,18) to (7,11)
through a series of open doors. [...]

Given the complexity, perhaps the first step is to go to the grey
door at (7,11), then open it, then go to the purple box. [...]

But without knowing the exact room layout, this is tricky. [...]

This is getting too vague.

Perhaps the best approach is to check for the presence of any doors
that the agent can pass through from their starting position [...]

So the subgoals would be:
1. (GoNextToSubgoal, (9,7)) - move to the green door.
2. (GoNextToSubgoal, (7,11)) - move to the grey door.
3. (OpenSubgoal) - open the grey door.
4. (GoNextToSubgoal, (5,16)) - go to the purple box.

But wait, after opening the grey door, does the agent need to
pass through it again? The purple box is at (5,16), which is
in the same room as the grey door [...] So after opening the door,
the agent can directly go to (5,16).
```

```
Therefore, the sequence of subgoals is as outlined.
</think>

<START>
(GoNextToSubgoal, (9, 7))
(GoNextToSubgoal, (7, 11))
(OpenSubgoal)
(GoNextToSubgoal, (5, 16))
<END>
```

The trace shows the model repeatedly second-guessing itself ("Wait", "But", "Alternatively"), exploring multiple redundant paths, and expressing uncertainty about basic environment mechanics, despite eventually arriving at a correct solution.

### I.3 QWEN FAILED REASONING TRACE EXAMPLE

The reasoning in Qwen's response exhibits repetitive self-correction and analysis paralysis:

```
<think>
Okay, let's see. The agent needs to reach the red ball at (15, 6).
The agent starts at (3, 16) facing west. [...]

Looking at the objects, there's a box at (1,16), a ball at (8,16)...
So moving north from (3,16) would be (3,15). Is there an object there?
The objects listed don't include anything at (3,15).
So maybe the agent can move north.

But the agent is facing west. To move north, the agent needs to turn
right (to face north) and then move forward. Let's outline the steps:
[...]

Wait, the user listed "key, color=grey, position=(3, 14)"...
So the agent can't move there. Therefore, the agent can't go
directly from (3,15) to (3,14). So the agent would have to find
another path.

Hmm, that complicates things. So moving north from (3,16) to
(3,14) is blocked by a key. So maybe the agent needs to move east
first, then north. [...]

Alternatively, the agent can turn right (to face north) and move
forward until it can't, then turn east. Let's see. From (3,16),
facing west, turn right to face north. Move forward to (3,15).
Then check if there's an object there. The objects at (3,15)
are not listed. Then move to (3,14), but there's a key at (3,14).
So the agent can't move there. Therefore, the agent must find
another way.

[... continues with similar patterns for 200+ lines ...]

This is getting complicated. Maybe there's a better way.
Let me look at the red ball's position again: (15,6). [...]

Hmm. This is tricky. Maybe the agent needs to find a different
path that avoids the key at (15,12) entirely. [...]

This is really challenging. Maybe I need to look for a different
approach. [...]

This is really frustrating. Maybe I need to consider that the
agent can't go directly and has to take a very long path. [...]

This is not working.
```

```
At this point, I'm stuck and need to find a different approach.
This is really challenging. Maybe the agent needs to take a very
long path that goes around all obstacles.
Alternatively, maybe
```

The trace demonstrates repetitive cycles of: (1) proposing a path, (2) discovering an obstacle, (3) expressing uncertainty ("Hmm", "Maybe", "Wait"), (4) abandoning the approach, and (5) restarting with a similar strategy, ultimately failing to converge on a solution.

