# OpenReview forum: "Through BabyAI Steps: Understanding and Evaluating Grounded Intelligence in LLMs"
_ICLR.cc/2026/Conference — Submitted to ICLR 2026_

### Official Review · Reviewer_SmRn · 2025-10-23

**Soundness:** 3
**Presentation:** 3
**Contribution:** 2
**Rating:** 4
**Confidence:** 4

**Summary:**

The paper introduces BABYBENCH, a textual adaptation of the BabyAI environment, to evaluate grounded reasoning in large language models through three components: predict, plan, and decompose. This benchmark aims to isolate core aspects of spatial reasoning and planning. The results show a clear gap between state prediction and goal-directed planning. The authors also test visual variants and find that visual grounding further degrades performance. This work provides a clean and reproducible benchmark for grounded reasoning. It's well executed but relatively narrow in scope and insights.

**Strengths:**

- The benchmark is well-structured and reproducible, with clear task definitions and metrics.
- The Predict-Plan-Decompose framework provides a simple and interpretable way to dissect different aspects of spatial reasoning.

**Weaknesses:**

- The paper’s contribution is mainly infrastructural; it does not introduce new modeling ideas or theoretical insights.
- The benchmark focuses on a very simplified, symbolic environment. The more interesting aspects of spatial reasoning lie in the visual information of shapes, sizes, directions, which are difficult to capture through text alone. The contribution of this paper would be more significant if the authors extend their study to visually described environments that involves more spatial structures.
- The evaluation lacks precise version information for API-based LLMs.

**Questions:**

In the current design, the "Decompose" task is evaluated through the OmniBot’s subgoal execution structure, where metrics depend on how well the bot executes or assists with the LLM-generated subgoals. However, this setup seems tightly coupled with the OmniBot’s own subgoal semantics and execution logic, which might limit the generalization of this metric to other tasks or environments. Have the authors considered a more “reference-free” approach for evaluating decomposition ability—one that relies on the LLM itself rather than an external executor? For example, could decomposition be assessed based on whether the generated subgoals lead to successful action sequences that fulfill these subgoals, and whether chaining these subgoals improves success on the original mission compared to direct planning without decomposition?

---

> ### Author Response · Authors · 2025-12-03
>
> We thank the reviewer for their positive feedback. We address the raised points below.
>
> **Regarding Weakness 1: Infrastructural vs. Modeling Contributions**
>
> While we agree that some of the primary contributions of the paper are infrastructural, we argue that the Predict-Plan-Decompose (PPD) framework offers a critical methodological contribution.
>
> - **Diagnostic Value:** Current modeling papers often struggle to diagnose why an agent fails in complex environments (Perception? Memory? Logic?). Our framework provides a standardized diagnostic harness. It allows researchers to pinpoint specific cognitive failures - for example, proving that a model's failure might not be due to a lack of "world dynamics" knowledge (Prediction success > 80%) but a failure of "search heuristics" (Planning success < 20%). We believe this diagnostic clarity is a prerequisite for the theoretical insights the field needs.
>
> **Regarding Weakness 2: Symbolic vs. Visual Environment**
>
> We deliberately chose a symbolic environment to isolate Grounded Reasoning from Perception.
>
> - **Symbolic Grounding is Critical:** Many real-world agentic tasks are purely symbolic yet spatially or structurally complex - for example, navigating file systems, manipulating DOM trees in web agents, or handling API dependencies. These domains require precise "spatial" manipulation of abstract states without visual cues.
>
> - **Isolating Logic:** In visually rich environments (like Habitat), reasoning failures are often confounded by noisy visual components. By using a noise-free text description, we isolate the reasoning engine. If SOTA models fail to reason about "shapes and directions" when they are perfectly described in text (as our results show), they are unlikely to succeed when those shapes must be inferred from noisy pixels.
>
> **Regarding Weakness 3: API Versions**
>
> Thank you for pointing this out. We apologize for this oversight. The precise version information for the API-based LLMs was documented in detail only within the paper-associated code repository. We will add a dedicated table in the Appendix listing the exact model snapshots (e.g., claude-sonnet-4-20250514) to ensure full reproducibility.
>
>
> **Question: Reference-Free Decomposition & OmniBot**
>
> Assessing the subgoal decomposition is harder than the two other tasks. If we assess it in a reference-free way (as is done with the precision rate metric), we fail to capture cases where a model is nearly correct, or where it only misses an anecdotal step of the process. The idea of using the OmniBot is that if the OmniBot needs to add many steps by itself, it indicates that the sequence of subgoals proposed by the LLM was far from correct. Conversely, if the OmniBot needs to add only one or two steps, it indicates that the LLM was nearly correct. This enables a more nuanced assessment, which is captured by the Assistance Curve Integral (ACI) metric.
>
> Moreover, the OmniBot is designed such that if the LLM has a basic understanding of how to reason in the BabyAI world and succeeds in providing a minimal decomposition that literally translates the goal into subgoals (e.g., if the task is “pick the red ball”, the LLM should at least return “Go To Red Ball”, “Pick Red Ball”), then the OmniBot will be able to add the missing intermediate steps to complete the task. When the OmniBot cannot do so, it indicates that the LLM lacks a basic understanding of how the world works. This is what is currently reflected by the Comprehension Rate (as reported above, we will change the name of this metric to “Assisted Success Rate” for more clarity).

---

### Official Review · Reviewer_n4uf · 2025-10-30

**Soundness:** 2
**Presentation:** 2
**Contribution:** 2
**Rating:** 2
**Confidence:** 4

**Summary:**

The contributions of the paper are two fold. First, they introduce BabyBench, a benchmarking environment and suite focused on evaluating the predictive reasoning and planning capabilities of LLMs within a randomly generated BabyAI grid world. Secondly, they provide a Predict-Plan-Decompose (which they name PPD) framework which is an evaluation framework whose goal is to isolate and evaluate atomic grounded reasoning abilities of the LLM models. Using this framework, they run a series of evaluations across multiple LLM models in order to uncover the relationship between predictive capabilities of these LLMs with long term planning capabilities. In doing so, they uncover an asymmetry between predictive and planning capabilities of LLMs.

**Strengths:**

Paper provides a new environments and evaluation suite for analysis of the predictive and planning capabilities of LLMs. Assuming that the code is open sourced, this artifact may be a valuable contribution to the field at large. Furthermore, much of the baselines and different configurations within the paper seem to be in anonymized github link, amplifying the potential for future work to build on top of the results here. For planning tasks in particular, it can be difficult to find benchmarks which can accurately assess the performance of these models clearly.

**Weaknesses:**

The main concern is that this predict-plan asymmetry derived as the main result of the analysis in the paper seems fairly obvious and covered in prior work. Multiple prior works have already explored the fact that planning is a particularly difficult task for LLMs in multiple different domains [1, 2, 3]. Furthermore, the idea that an LLM can act as a form of a world model to predict the future has also been covered in prior work. [3, 4] So the main "notable dissociation" derived in this work seems to be well covered in prior work. Abstractly, if a model achieves per-step prediction accuracy $p \in (0,1)$, the success probability for $n$-step planning is bounded by $p^n$, which decreases exponentially as $p^n < p$ for all $n \geq 2$. Furthermore, this kind of derogation could also due to a longer-context, since the predict task only requires the model to predict the final state, whereas the plan and decompose task requires the model to synthesize multiple steps.

In addition, as mentioned in Section 5.1:

Predict does not require the LLM to form a faithful spatial internal representation. In practice, models that perform well on this task implement a shallow arithmetic routine rather than reasoning about the environment. As a result, the model can ignore the spatial context and rely solely on integer additions. [Lines 412-415]

It seems like then, the high performance of these models on the Predict task can be partially attributed to the Grid-like structure of the benchmark, where the model can do "a shallow arithmetic routine" to predict the next step. This may explain why the Predict score is so high. For a non-grid world task, one might expect the models to do more poorly on a predict task, where the models cannot exploit this.

Furthermore, the experiments are not extensive enough to derive this result robustly. The reasoning for choosing Tree-of-thoughts seems fairly barebones. Appendix E and Table 7 show that the ToT strategy does not perform that much stronger compared to Chain-of-thoughts (CoT) or Few-Shot, and further experiments using these prompting strategies would be strengthen the results shown in the paper. Similarly, the justification for using the Structured format in Section 3.2 is also lacking.

Overall, if one could show that these results across models hold across different prompting strategies and output formats, this could lend stronger insight into the abilities of these different LLMs.


Presentation-wise, the Figures in the paper are extremely difficult to read and analyze. Figures 2 and 3 and Table 3 have extremely small font and axes which make it hard to interpret the results. There are also several typos across the paper, for example, the title of Section 5.1 (Assymetry -> Asymmetry).

[1] Valmeekam, Karthik, et al. "Planbench: An extensible benchmark for evaluating large language models on planning and reasoning about change." Advances in Neural Information Processing Systems 36 (2023): 38975-38987.

[2] Webb, Taylor, Shanka Subhra Mondal, and Ida Momennejad. "Improving planning with large language models: A modular agentic architecture." arXiv preprint arXiv:2310.00194 (2023).

[3] Hao, Shibo, et al. "Reasoning with language model is planning with world model." arXiv preprint arXiv:2305.14992 (2023).

[4] Gu, Yu, et al. "Is your llm secretly a world model of the internet? model-based planning for web agents." arXiv preprint arXiv:2411.06559 (2024).

**Questions:**

The main suggestion to the authors would be to provide evidence for this predict-plan asymmetry existing independently from the quirks of the benchmark or from the long-context concerns. For instance, what is the average prompt length for the model and generation lengths between the predict, plan, and decompose parts of the benchmark? If the number of tokens used across the board is similar, than this concern can be at least partially mitigated. How deep is the ToT depth on average across the board? An experiment using something simpler like CoT or Few-Shot would make this kind of analysis significantly simpler and provide stronger evidence for the main result.

Another question is about Section 5.3, where it states that the vision model yielded worse results than the LLM counter part. This section is incredibly barebones, as it does not even have a single reference, table, or figure attributed to it. Did the GPT5 vision model achieve a 0% accuracy across the board? Was the same prompt used between the LLM GPT5 and the VLM GPT5? This seems more like a prompting strategy error rather than a fundamental flaw in the VLM.

---

> ### Author Response · Authors · 2025-12-03
>
> We thank the reviewer for their detailed critique. We address your concerns and questions below.
>
> ## **Weaknesses**
>
> **Regarding Weakness 1: Novelty of Predict-Plan Asymmetry**
>
> While prior works ([1], [2]) have established that planning is a difficult task for LLMs in multiple different domains, and others ([3], [4]) have explored LLMs as world models, our specific contribution is the quantitative isolation of the dissociation between these two capabilities in a controlled setting.
>
> The common hypothesis in Model-Based RL is that "a better world model leads to better planning". Our results challenge the "naive" application of this to LLMs. We show that a model can have a near-perfect "world model" (Prediction success ~ 95%) yet fail completely at (high/low level) planning (Planning success ~ 15%).
>
> You noted that planning success $p^n$ decreases exponentially. While true for random errors, our tested models fail catastrophically even on short horizons (small $n$) where $0.95^n$ would still predict success. This suggests the failure isn't just cumulative error, but a functional disconnect: the model cannot "invert" its predictive capability to form a search heuristic.
>
> **Regarding Weakness 2: "Shallow Arithmetic" and Grid Artifacts**
>
> We argue this is a feature, not a bug, for diagnosing reasoning depths.
>
> The fact that models can exploit this arithmetic for Prediction but cannot reverse-engineer it for Planning is the central insight. If the reasoning were robust (i.e., constructing a mental map), the inverse operation (finding a path) should be accessible, at least for sparse grids. The fact that it isn't proves that the "reasoning" is merely local pattern matching (next token prediction) rather than global state simulation.
>
> In non-grid tasks (e.g., continuous control), models would likely fail Predict as well. By using a grid, we show that even when Predict is solved via a "hack" (arithmetic), Plan remains unsolved.
>
> **Regarding Weakness 3: Experimental Extensiveness (ToT vs. CoT)**
>
> We agree that showing robustness across prompting strategies strengthens the claim. We performed a sweep of strategies (Zero-shot, Few-shot, CoT, ToT). We found that while ToT offered marginal quantitative gains on easy tasks, it was qualitatively better at preventing "infinite loops" in navigation.
>
> **Regarding Weakness 4: Presentation (Figures & Typos)**
>
> We apologize for the readability issues.
>
> - **Action:** We will regenerate Figures 2 and 3 with larger fonts and thicker axes. We will also fix any eventual typos and perform a full proofread.
>
>
>
> ## **Questions:**
>
> **Question 1: Context Length & Evidence for Asymmetry**
>
> We analyzed the token counts to rule this out:
>
> - **Context & Generation Lengths:** The input context length is almost identical for Predict and Plan/Decompose. For a task, Plan/Decompose produces a sequence of steps (actions or subgoals), while Predict outputs incremental coordinate updates. Predict’s outputs are more compact, but the challenge shifts to maintaining and updating the model’s internal state.
>
> - **Evidence:** The same asymmetry persists even when the underlying task requires only a few steps to solve - situations where extra context is irrelevant or even distractive. Yet Plan and Decompose still degrade far more than Predict. This shows that the issue isn’t driven by context length but by something fundamental to the planning-style generation itself.
>
> **Question 2: Vision Results (0% Accuracy)**
>
> We would like to clarify the following:
>
> - **Result:** The model indeed achieved 0% accuracy on the Plan task when relying on the visual modality.
>
> - **Why:** This was not a prompting error (we used the same optimized prompt structure). The failure arose from a modality-specific limitation: in the VLM condition the model received only the screenshot plus the fixed rules preamble, unlike the LLM which received explicit symbolic coordinates. GPT5-Vision therefore relied on coarse, region-level spatial cues (“lower right area”) rather than inferring precise $8 \times 8$ grid coordinates from pixels (“cell (6, 6)”). These small localization errors compounded across steps because the task requires stable symbolic state tracking for long-horizon planning. As a result, the model frequently misaligned objects on the grid and produced invalid or misdirected actions. We present this as specific to GPT5-Vision in this setup, not a general claim about multimodality.

---

### Official Review · Reviewer_EiLs · 2025-11-01

**Soundness:** 3
**Presentation:** 2
**Contribution:** 2
**Rating:** 4
**Confidence:** 3

**Summary:**

The paper introduces BABYBENCH, a text-based benchmark built from the BabyAI gridworld to test LLMs’ grounded reasoning and planning. It defines three tasks — predicting action outcomes, planning action sequences to reach goals, and decomposing high-level instructions into subgoals.

Experiments show that while top models (GPT-5, Claude 4, LLaMA 3.1, Qwen 3, etc.) perform well on short-term prediction (>80% accuracy), they fail on long-horizon planning (<20% success), with performance dropping further (~10–12%) under partial observability and interactive settings. Also, multimodal models perform even worse when given visual inputs.

Overall, BABYBENCH offers a clean, procedurally generated testbed revealing that current LLMs can simulate short-term effects but lack robust spatial and grounded reasoning for long-term planning.

**Strengths:**

Problem is important, benchmark is helpful:
The paper tackles a core limitation in current LLM research — the lack of true grounded reasoning and planning abilities. By introducing BABYBENCH, a controlled text-based gridworld benchmark, the authors isolate reasoning from visual and linguistic noise seen in prior embodied tasks. This lets them probe core spatial and planning competence directly. The benchmark sits neatly between abstract text-based planning datasets (e.g., NATURAL PLAN) and complex embodied simulators (e.g., ALFRED, ALFWorld), filling an important methodological gap.

Comprehensive evaluation experiments done:
The study tests multiple leading LLMs across architectures and scales under standardized settings, ensuring generality of the findings. Chain-of-Thought, Tree-of-Thought prompting are applied to give models the best chance to succeed, and an expert BabyAI OmniBot agent supplies gold-standard plans and automated execution. The graded difficulty levels (easy to very hard) help characterize how performance deteriorates with increasing task complexity, a crucial insight into long-horizon reasoning.

Reproducibility and Open Science:
BABYBENCH is fully open-source and procedurally generated, ensuring virtually infinite data diversity and eliminating contamination. The authors release the codebase, dataset variants, and environment agents, making replication and extension straightforward. This would potentially be a major contribution to the field and promotes long-term spacial grounded reasoning across models and approaches.

**Weaknesses:**

Visual degradation is preliminary conclustion:
The finding that visual input worsens performance is intriguing but based on only one multimodal model (GPT-5). It’s unclear whether the failure is from inherent limitations of vision-language reasoning or from that model’s specific training and representation biases. Broader testing, with GPT-4 Vision, Gemini, or with vision representation training would help validate it. The vision result should be treated as an early observation rather than a general conclusion. Including even one more baseline or a controlled text-vs-image comparison would strengthen this part of the study. Research such as Jin, Emily, et al. "MARPLE: A benchmark for long-horizon inference." shows contradictary multi-modality improvement for long horizon reasoning.

Synthetic Setting and Transferability is unclear:
The BabyBench environment’s minimalist “Baby Language” and gridworld setup enable clean analysis but raise questions about generalization. While the benchmark isolates core reasoning, it diverges from natural, noisy, or visually rich environments. The authors acknowledge this and relate it to prior work (e.g., NATURAL PLAN’s similarly low success rates), but it remains unclear how much BabyBench results predict real-world embodied reasoning. The benchmark’s artificial precision may also encourage strategies that would not transfer, such as coordinate arithmetic. A short discussion of how findings might extrapolate to messier inputs or language-based control would provide useful context.

Incompleteness of Partial Observability Experiments:
The FPI (interactive) evaluation is an excellent addition but limited in scope, only two models were tested, and only on a small subset of missions. This constrains what we can infer about architecture-specific behaviors under feedback and memory constraints. The results (10–12% success) convincingly show difficulty, but more detailed analysis, such as whether failures stem from looping, goal forgetting, or perceptual confusion would add depth. The paper briefly mentions efficiency metrics but doesn’t interpret them, missing a chance to reveal whether models are inefficiently correct or simply fail altogether. These are minor but notable omissions in an otherwise rigorous evaluation.

Minor Clarity Issues:
The writing is clear overall, but a few details could improve readability. The custom metrics for the Decompose task are only defined in the appendix and would benefit from a short intuitive description in the main text. Similarly, quantitative comparisons among prompt formats (structured vs narrative or JSON) would help contextualize the chosen setup. Lastly, the label “GPT-5” for the multimodal model may confuse some readers; clarifying it as a future or prototype GPT variant would avoid ambiguity. These are small, easily fixable issues that do not detract from the paper’s overall quality.

**Questions:**

Potential for Training/Fine-tuning: Did you try fine-tuning an LLM (or training a smaller model) on BabyBench tasks to see if planning improves? The current study focuses on zero/few-shot results, but it would be helpful to know if the poor planning is due to lack of exposure or deeper architectural limits. Even a brief discussion or intuition on whether fine-tuning could bridge the gap would add valuable perspective.

Vision Input and Failures: How exactly was the visual input provided impact the outcome: as a rendered grid image, textual description, or something else? And what kinds of mistakes did the model make, did it misread spatial layouts or simply ignore the visual cue? Since only GPT-5 was tested, it’s unclear if this reflects a general weakness of VLMs or that specific model. Clarifying whether the issue is coarse spatial representation or domain mismatch (natural vs schematic images) would strengthen this point.

Long-Horizon Planning Behavior:
In the Tree-of-Thought setup, did models actually branch and explore multiple plan paths, or were they mostly greedy? Examples of near-misses where the model almost solved a task but made a wrong move, would help illustrate whether the failure is due to poor search, limited memory, or state-space complexity.

Scaling with Environment Size:
As grid size and obstacle count increased, did you observe any behavioral trends, e.g., overly long, nonsensical plans or premature halts? Insights into whether errors stem from sequence length (depth) or environment clutter (breadth) could clarify what truly limits scaling.

---

> ### Author Response · Authors · 2025-12-03
>
> We thank the reviewer for the thoughtful assessment. We address their concerns and questions below.
>
> ## **Weaknesses**
>
> **Regarding Weakness 1: On the Vision-Based Condition and Generalization**
>
> To clarify the setup: in the VLM condition, the model received only a screenshot of the initial environment state, along with the same fixed textual preamble used in all conditions (explaining rules, actions, coordinates, etc.). It did not receive the textual environment description provided to the LLM. The comparison between GPT5 ("LLM") and GPT5-Vision ("VLM") is therefore a clean text-only vs. image-only modality contrast.
>
> Under this setting, GPT5-Vision consistently underperformed. Our analysis explains this model-specific behavior: the VLM relied on coarse, region-level spatial cues (e.g., “The red ball is in the lower right area of the grid.”) and did not reliably infer precise grid coordinates from pixels (e.g., “The red ball is located at cell (6, 6).”), yielding localization errors and failures in long-horizon state tracking. In contrast, the text-based model, given explicit symbolic coordinates, exhibited more discrete and stable planning. We thus present this finding as specific to this model and modality, not as a general claim about multimodality.
>
> Regarding MARPLE, we believe the discrepancy arises from task demands: MARPLE benefits from semantic visual recognition, whereas BabyBench requires exact coordinate alignment, where even a single-cell error can cause failure.
>
> Finally, GPT5-Vision was chosen because, among the multimodal models we explored (e.g., Gemini-2.5-Pro, Qwen-2.5-VL), it offered greater potential and enabled a more controlled LLM vs. VLM comparison.
>
> **Regarding Weakness 2: Synthetic Setting & Transferability**
>
> We argue that the synthetic nature of BabyBench serves as a necessary "Sanity Check". If SOTA models fail to plan in a noise-free, fully observed gridworld, where the rules are deterministic, they are unlikely to succeed in messy, real-world embodied environments via reasoning alone. We agree that "coordinate arithmetic" is an artifact of the gridworld. However, our results show that models cannot even transfer this arithmetic capability from Prediction to Planning. This failure to generalize a simple arithmetic rule to a search heuristic is a vital insight into the limitations of current LLM "reasoning".
>
> **Regarding Weakness 3: FPI Analysis**
>
> We agree that the FPI section would benefit from deeper behavioral analysis.
>
> - **Action:** Our preliminary analysis shows that failures in FPI are dominated by "State-Estimation Drift" (forgetting the map outside the current view) rather than looping. We will expand Section 4.3 to better categorize failure modes.
>
> **Regarding Weakness 4: Clarity (Metrics & GPT-5 Label)**
>
> - **Decompose Metrics:** We will move the definitions of “Assistance Curve Integral” and “Precision Rate” to the main text and rename “Comprehension Rate” to "Assisted Success Rate" for clarity.
> - **Prompt Formats:** As suggested, we will add a short quantitative comparison of Structured vs. Narrative/JSON prompt formats to the main text to better contextualize our chosen setup.
> - **GPT-5 Label:** To avoid any confusion, we will use the exact model API names in the final paper.

---

> > ### Author Response · Authors · 2025-12-03
> >
> > ## **Questions**
> >
> > **Question 1: Potential for Fine-tuning**
> >
> > We did not perform fine-tuning in this study, focusing on exploring the potential of different sophisticated prompting strategies. We hypothesize that fine-tuning would readily solve the Predict task (learning the transition function $S_t \times A_t \to S_{t+1}$). However, we suspect it would have diminishing returns on the Plan & Decompose tasks for unseen grid settings. The "shallow arithmetic" we observed suggests fine-tuned models might overfit to a specific type of settings (e.g., "memorizing" 8x8 paths) rather than learning the Dijkstra/A* logic required for general pathfinding (which the OmniBot approximates in the form of BFS-based shortest-path search). We will add this discussion to the Future Work section.
> >
> > **Question 2: Vision Input Mechanics**
> >
> > - **Input:** In addition to a textual preamble (explaining the rules, the actions, the coordinate system, etc), the visual input was a high-resolution rendered image of the environment initial state.
> > - **Mistakes:** The primary error was Visual-Symbolic Misalignment. The model would correctly see "The red ball is in the bottom right" (visual cue) but generate an action sequence doomed to failure, resulting in a final state completely different from the one desired. The text-only tests forced the models to rely on the explicit coordinates, and made fewer of these imprecision errors.
> >
> > **Question 3: ToT Behavior**
> >
> > In our ToT experiments, models did branch, but the bottleneck was the Self-Evaluation step. Models generated diverse candidate continuations, but when selecting among them, the search tree usually gets pruned incorrectly (despite non-greedy generation) due to some of the following failure modes:
> >
> > - **Memory / state-tracking:** models sometimes confused the initial environment with the evolving hypothetical states, leading to coordinate drift and inconsistent internal maps.
> >
> > - **Poor search initialization:** some branches were poorly grounded from the outset (due to mislocalized agent or goal), making entire trajectories effectively doomed from the first expansion.
> >
> > - **State-space complexity:** success rates dropped markedly in environments with more rooms, obstacles, or objects, indicating difficulty scaling ToT reasoning to larger combinatorial spaces.
> >
> > **Question 4: Scaling (Depth vs. Breadth)**
> >
> > We observed distinct failure modes depending on how the environment was scaled:
> >
> > - **Grid Size (Depth):** As grid size increased, models produced increasingly nonsensical plans later in the sequence. They often lost track of the agent’s virtual position after many simulated steps, leading to “hallucinated geometry” (e.g., attempting to turn a corner already passed or steering into empty space).
> >
> > - **Obstacle Count (Breadth):** With more obstacles, errors arose much earlier: models committed to locally valid but globally dead-end paths. These failures arise from the model struggling to choose among many plausible paths when the environment becomes cluttered.

---

### Official Review · Reviewer_NKXr · 2025-11-01

**Soundness:** 2
**Presentation:** 3
**Contribution:** 2
**Rating:** 4
**Confidence:** 4

**Summary:**

This paper uses the BabyAI gridworld to evaluate the prediction, planning, and decomposition ability of LLMs. Prediction is the ability to map a current state and sequence of actions to the future state, planning is the ability to map a state and subgoal to a sequence of actions, and decomposition is defined as the ability to map instructions into sequences of subgoals. The results show that LLMs are fairly good at prediction, but bad at planning and decomposition.

**Strengths:**

1. Originality: As the paper acknowledges, there is a large body of work studying the planning/decomposition abilities of LLMs and a large body of work trying to make pretrained models better dynamics models. To me, the main contribution of this work is that it studies the different aspects of sequential decision making--forward models (prediction), higher-level planning (decomposition), and lower-level planning (planning).
2. Quality: The work seems comprehensive, with open-sourced code and environments within BabyAI. There are some interesting experiments in the appendix.
3. Clarity: The work is easy to read and understand.
4. Significance: This paper can conceivably form part of a benchmark to better understand and iterate on LLM sequential decision making capabilities along the three aspects of predict-plan-decompose.

**Weaknesses:**

1. While the work proposes three axes along which to evaluate LLM sequential decision making capabilities, it does not propose any insights about how to potentially improve their capabilities along these fronts.
2. Most of the core findings seem too specific to BabyAI gridworlds and not as applicable to LLMs on sequential decision making problems in general (Sections 5.1 - 5.3).
3. Table 1 proposed metrics like manhattan distance that were not reported in subsequent results sections. (If the metric only shows up in the appendix, then it should not be in the main text.)
4. Lines 275-281 mentions 4 different prompting strategies but Line 283 mentions all experiments use Tree-of-Thought (ToT). I would recommend authors to simplify this paragraph and just note ToT is used, without mentioning the other prompting strategies, to simplify and improve clarity.
5. In general, in my opinion, there are too many metrics (7) in the main text of the paper, and I would suggest the authors focus on a smaller group of metrics, with more analysis on each. Some metrics like comprehension rate are confusingly named, and most of them are not adequately explained or motivated.
6. Line 412-414: “Models that perform well on [Predict] implement a shallow arithmetic routine rather than reason about the environment.” Was this conclusion based on an analysis of LLM outputs? Please provide some concrete metrics. Also, arithmetic on the (x,y) positions could be an important part of environment reasoning. How generalizable are these lessons outside of gridworld?
7. Line 415-416: “LLMs that failed on Predict tended to over-engage with the spatial context, for example by validating each action, checking the prompt, or questioning feasibility.” Why is it bad to “over-engage,” “validate actions,” “check the prompt,” and “question feasibility”?
8. Lines 428-433. Instead of listing the failure modes, count their frequency, categorize them, and show them as a pie chart.
9. Section 5.3 seems kind of weak due to the lack of concrete results. Authors say there are no experimental results because “they yielded zero outcomes.” However, is this true even in the easy 4x4 grids? And it was zero success rate over how many trials?

**Questions:**

1. One of the main questions that comes to mind after reading about the results -- can LLMs do better planning by leveraging their strength on the prediction problem? Basically, have the prompt encourage an intermediate prediction step before it outputs actions for planning. If authors can demonstrate some performance gains by doing this, it would make this a much more exciting and strong paper, in my opinion.
2. What does “assistance curve integral” intuitively mean? The word “assistance” is confusing. What is assisting what? What module is proposing additional LLM subgoals, and how does this proposing occur? Why is this metric important?
3. Section 5.3: What happens if the image is provided alongside the same textual state description? Is the success rate still zero? I think authors would benefit by reporting results for three modality scenarios: text-only state description, text + image state, and image-only. One can also try a slightly modified image state where the rows and columns are explicitly numbered, to facilitate coordinate understanding.
4. Authors say in Section 5.3 that the VLM is much less precise than LLMs; the VLM uses words like “make a short south adjustment” while the LLM says things like “the red ball is at (4, 14).” How much of this is due to the fact that the LLM was given a list of precise coordinates for all objects on the scene, versus the VLM which was presumably not given any coordinates?
5. What are some findings that you talked about in Section 5 that are applicable to an LLM in a continuous state-space setting?
6. Section 4.3: Why should we expect that an agent with partial observability can perform decent planning? What kind of -- and how much -- interactive execution are they allowed?

---

> ### Author Response · Authors · 2025-12-03
>
> We thank the reviewer for their feedback. We address their specific concerns and questions below.
>
> ## **Weaknesses**
>
> **Regarding Weakness 1: Insights on improving capabilities**
>
> We agree that diagnosing the problem is only the first step. While the suggested modification (in Question 1) adds an explicit prediction step to the planning prompt, our existing prompting strategy already encourages analogous behavior: the ToT prompt promotes consequence-based reasoning, and the Predict prompt teaches state-transition modeling. This implicitly captures the intended effect of the proposed intervention. We elaborate on this point in our response to Question 1.
>
> **Regarding Weakness 2: Specificity to BabyAI Gridworlds**
>
> While BabyAI is a gridworld, we argue that the "shallow arithmetic" vs. "grounded reasoning/planning" distinction is best observed here. In complex, "realistic" simulators (like Habitat or ALFRED), it is difficult to disentangle whether a failure is due to visual perception noise or reasoning failures. By using a controlled text-based grid, we isolate the reasoning component. The finding that models can track state changes (arithmetic) but cannot chain them into a goal-directed policy is likely applicable to any domain requiring multi-step tool use or API interaction (e.g., coding agents), where "knowing what a function does" (Predict) is different from "knowing which function to call to solve a problem" (Plan/Decompose).
>
> **Regarding Weakness 3 & 5: Metrics (Manhattan Distance, Comprehension Rate, too many metrics)**
>
> We agree that the number of metrics can be streamlined for clarity.
>
> - **Action:** We will move Manhattan Distance and secondary metrics to the Appendix. The main text will focus strictly on Success Rate and Efficiency.
>
> - **Clarification:** We will rename "Comprehension Rate" to "Assisted Success Rate" to be more intuitive. Unlike “Precision Rate” (which reports the accuracy with no external help), it measures whether the model understood the high-level intent of the task, even if it required the OmniBot to inject intermediate subgoals to fix execution errors.
>
> **Regarding Weakness 4: Prompting Strategies**
>
> We agree this could be avoided for more clarity.
>
> - **Action:** We will simplify the methodology section to state that Tree-of-Thought (ToT) is our primary evaluation method. We will move the comparison of prompting strategies (Zero-shot vs. Few-shot vs. CoT vs. ToT) to the Appendix to improve flow and clarity.
>
> **Regarding Weakness 6: "Shallow Arithmetic" Conclusion**
>
> This conclusion is based on a qualitative error analysis. We observed that in the Predict task, models, for example, would often correctly predict (x+1, y) for a “forward” action, even when (x+1, y) contained a wall (which should block the move). This indicates the model is performing integer addition on the coordinates provided in the text, rather than reasoning about the spatial constraints of the environment.
>
> When it comes to generalizability, this suggests that outside of gridworlds, LLMs may excel at tasks that look like "pattern matching" or "syntactic transformation" (arithmetic) but fail at tasks requiring "simulation of constraints" (physics/logic), a finding consistent with recent work on LLMs in planning.
>
> **Regarding Weakness 7: "Over-engaging"**
>
> Excuse the ambiguity. By "over-engaging," we meant that models often hallucinated complexity that did not exist. For example, models would stop to question the validity of the task itself, despite the prompt explicitly stating the game rules. This behavior was "bad" in this context because it led to the model refusing to generate valid actions for solvable tasks. We will clarify this in the main text.
>
> **Regarding Weakness 8: Pie Chart for Failure Modes**
>
> This is an excellent suggestion to improve the presentation of our qualitative analysis.
>
> - **Action:** We will categorize the failure modes (e.g., "Wall Collision," "Infinite Loop," "Hallucinated Obstacle," "Syntax Error") and present them as a pie chart in the final version.
>
> **Regarding Weakness 9: Section 5.3 (Vision Results)**
>
> We clarify that the 0% success rate was observed across multiple  difficulty levels, including the Easy (8x8) grids. The failure was fundamental: the VLM could consistently fail to map pixels to the precise discrete grid coordinates required to output a valid action plan (e.g., identifying the ball at (3,4) when it was actually at (3,5)), leading, for example, to immediate wall collisions.

---

> > ### Author Response · Authors · 2025-12-03
> >
> > ## **Questions**
> >
> > **Question 1: Prediction-Guided Planning**
> >
> > Our ToT prompting already partially tests this idea: during planning it encourages step-by-step reasoning about goals, paths, and obstacles, and in our separate Predict prompts it explicitly simulates state transitions. While we don’t explicitly enforce a strict “predict-then-act” substep at every planning turn, the existing setup already pushes the model to reason locally about consequences, capturing the central spirit of the proposed intervention. A dedicated variant that forces an intermediate next-state prediction before each action would therefore be a more explicit version of behavior the model is already encouraged to perform under our current prompting scheme.
> >
> > **Question 2: Assistance Curve Integral (ACI)**
> >
> > - **Intuition:** ACI can be seen as Robustness to Partial Failure.
> > - **Who assists?** The OmniBot (expert agent) acts as a safety net. If the LLM outputs a plan with a gap (e.g., forgets to open a door), the OmniBot inserts the necessary subgoals to keep the run going. In this context, the ACI is the integral of the curve rate of success as a function of assistance needed (this is why we called it the Assistance Curve Integral).
> > - **Why it matters:** A binary Success/Fail metric ignores "mostly correct" plans. ACI distinguishes between a model that missed 1 step (high ACI) and a model that hallucinated entirely (low ACI).
> >
> > **Question 3: Image + Text & Modality**
> >
> > Our rationale is that, given the way the environment is specified, the textual description already provides a complete and unambiguous encoding of the grid (exact coordinates, obstacle positions, agent location, goal location, etc). In this setting, the model does not need to extract any information from the image, and adding the screenshot would not meaningfully test multimodal integration. Instead, it would almost certainly lead the model to default to its stronger textual reasoning ability, masking whether it can actually interpret and align visual spatial cues.
> >
> > For this reason, a text + image condition would not reveal genuine multimodal understanding, but rather whether the model prefers one modality when both are redundant. The image-only condition is therefore the more diagnostic setup for evaluating the model’s capacity for visual spatial reasoning. Likewise, modifying the image to include row/column indices would simply reintroduce textual structure visually, without changing the fundamental issue that the model already receives a perfectly structured coordinate representation in text.
> >
> > **Question 4: VLM precision vs. Coordinates**
> >
> > The LLM succeeds largely because it is spoon-fed symbolic coordinates, whereas the VLM must infer them from pixels. This comparison is intentional: it shows that the LLM’s “spatial reasoning” is actually symbolic processing, while the VLM struggles to convert visual input into the precise discrete coordinates required for reliable actuation. Aside from this difference in state representation, both models receive the same rules preamble, so the gap reflects a representational bottleneck rather than unequal task framing. We emphasize that this is a model-specific observation rather than a general claim about all VLMs.
> >
> > **Question 5: Continuous State-Space**
> >
> > Our findings suggest that if LLMs struggle to chain causal dependencies in a discrete, simplified grid, they will likely struggle more in continuous spaces where the "arithmetic" hack (x+1) is unavailable and the causal gradients are finer.
> >
> > **Question 6: Partial Observability Planning**
> >
> > We expect agents to perform decent planning under partial observability by maintaining an internal state (mental map) of explored areas. They are allowed step-by-step interactive execution up to an appropriate limit that scales with the grid size. The failure of strong models here implies an inability to maintain state consistency over long contexts.

---

### Author Response · Authors · 2025-12-04
**Final Comment**

We thank the reviewers and the ACs for the careful evaluation. Our main goals with this work is to provide BabyBench (a BabyAI-based benchmark for LLMs/LMMs) and the diagnostic Predict-Plan-Decompose (PPD) framework to probe three aspects of grounded intelligence in a controlled setting: predicting state transitions, generating goal-directed action sequences, and decomposing high-level instructions into subgoals. Across several strong models, we consistently see high performance on spatial prediction (typically >80%) but much weaker performance on multi-step planning (<20%), with success dropping to ~10% in the Full-mission, Partial-observability, Interactive (FPI) setting.

Our claim is not merely that “planning is hard for LLMs” (which is known), nor that “LLMs may be able to act as world models” (also known). The key contribution is to co-locate these two capabilities in a single controlled testbed and to quantitatively demonstrate a sharp dissociation under matched conditions, showing that a model can have a near-perfect approximation to the transition function $S_t \times A_t \to S_{t+1}$ while still failing badly at planning. In particular, the usual bound that $n$-step planning success is at most $p^n$ for per-step accuracy ($p$) cannot fully explain the observed failures, because models already collapse on problems with short horizons (small $n$) for which $p^n$ would still predict high success. In model-based RL a common working hypothesis is that “a better world model leads to better planning”; our results challenge the “naive” application of this idea to LLMs.

This is reinforced by the structure of the Predict regime: world-modeling is about as easy as it can get, since the model can rely on a trivial $(x, y) \to (x \pm 1, y \pm 1)$ arithmetic routine on symbolic coordinates. Yet the same models struggle to invert this capability when asked to find action sequences (locally applying $S_t \times A_t \to S_{t+1}$ but not reliably solving the inverse problem of reaching a target state) suggesting that the “world model” they implicitly learn is used as local pattern matching rather than as a global planning substrate, and that if they cannot chain these simple symbolic transitions into coherent plans here, they are likely to struggle even more in continuous or visually rich domains where the “arithmetic hack” is unavailable.

We also clarified the points raised in the discussion and will reflect them in the final version. In particular, we will simplify and better explain the evaluation metrics, tighten and reorganize the exposition of prompting strategies, provide a clearer analysis of failure modes in the FPI and visual settings, and improve reproducibility and presentation details (including model specification, figures, and typos).

We hope this helps see what our work is actually pinning down: even in a deliberately simple, deterministic setting where Prediction is nearly solved, current LLMs exhibit a pronounced disconnect between accurate short-term dynamics and reliable high/low level Planning. BabyBench and the PPD framework make this disconnect measurable and comparable across models, and we see them as tools the community can use to diagnose and eventually close that gap.

---

### Meta-Review · Area_Chair_iTLr · 2026-01-04

**Summary:**

The paper introduces BabyBench, a text-based benchmark built from the BabyAI gridworld to test LLMs’ grounded reasoning and planning. It defines three tasks — predicting action outcomes, planning action sequences to reach goals, and decomposing high-level instructions into subgoals. Using this framework, they run a series of evaluations across multiple LLM models in order to uncover the relationship between predictive capabilities of these LLMs with long term planning capabilities.

**Reviewer Concerns:**

The reviewers' concern is about the lack of insights and analysis, as well as the unclear synthetic setting and transferability, and the incompleteness of Partial Observability Experiments.

The authors provide additional clarification, but it is not enough to address the concerns.

**Reviewer Scores:**

The reviewers' scores are consistently below the acceptance threshold.

---

### Decision · Program_Chairs · 2026-01-26

Reject